# Phytotoxin production in *Aspergillus terreus* is regulated by independent environmental signals

Markus Gressler[1], Florian Meyer[2], Daniel Heine[2], Peter Hortschansky[3], Christian Hertweck[2], Matthias Brock[4,5]*[†]

[1]Microbial Biochemistry and Physiology, Leibniz Institute for Natural Product Research and Infection Biology, Hans Knoell Institute, Jena, Germany; [2]Biomolecular Chemistry, Leibniz Institute for Natural Product Research and Infection Biology, Hans Knoell Institute, Jena, Germany; [3]Molecular and Applied Microbiology, Leibniz Institute for Natural Product Research and Infection Biology, Hans Knoell Institute, Jena, Germany; [4]Institute for Microbiology, Friedrich Schiller University, Jena, Germany; [5]Fungal Genetics and Biology Group, School of Life Sciences, University of Nottingham, Nottingham, United Kingdom

*For correspondence: matthias. brock@nottingham.ac.uk

Present address: [†]Microbial Biochemistry and Physiology, Leibniz Institute for Natural Product Research and Infection Biology, Hans Knoell Institute, Jena, Germany

Competing interests: The authors declare that no competing interests exist.

**Abstract** Secondary metabolites have a great potential as pharmaceuticals, but there are only a few examples where regulation of gene cluster expression has been correlated with ecological and physiological relevance for the producer. Here, signals, mediators, and biological effects of terrein production were studied in the fungus *Aspergillus terreus* to elucidate the contribution of terrein to ecological competition. Terrein causes fruit surface lesions and inhibits plant seed germination. Additionally, terrein is moderately antifungal and reduces ferric iron, thereby supporting growth of *A. terreus* under iron starvation. In accordance, the lack of nitrogen or iron or elevated methionine levels induced terrein production and was dependent on either the nitrogen response regulators AreA and AtfA or the iron response regulator HapX. Independent signal transduction allows complex sensing of the environment and, combined with its broad spectrum of biological activities, terrein provides a prominent example of adapted secondary metabolite production in response to environmental competition.

## Introduction

Fungal secondary metabolites (SMs) are often encoded by gene clusters and have been the subject of intensive analysis in recent years. This has resulted from the large number of completed fungal genomes as well as the intrinsic interest of SMs as environmental/pathogenic agents and their potential for pharmacological use. However, there are few examples where the regulation of the expression of these clusters has been shown to be ecologically/physiologically relevant (*Rohlfs and Churchill, 2011*). Except for penicillin and perhaps aflatoxin and gliotoxin, the potential real-life functions of most SMs are unknown (*Bhatnagar et al., 2006*; *Vargas et al., 2014*). Most clusters contain a pathway-specific transcription factor, and the common approach is to artificially over-express the cluster genes by constitutive expression of this transcription factor and chemically characterise the product (*Yin and Keller, 2011*). In most cases the relevance of the SM is not known.

In this study the biological function of terrein from the filamentous ascomycete *Aspergillus terreus* was investigated. *A. terreus* is a filamentous ascomycete of biotechnological and medical importance since it produces itaconic acid (*Steiger et al., 2013*) and lovastatin (*Bizukojc and Ledakowicz, 2015*),

**eLife digest** Organisms produce a wide variety of small molecules called metabolites through the break down of food and other chemical reactions. Some of these molecules—known as primary metabolites—are required for growth, reproduction and other vital processes. Other molecules called secondary metabolites are not strictly required by the organism, but generally have other roles that may improve the individual's ability to survive and reproduce.

Fungi and other microbes produce a large variety of secondary metabolites, many of which are used as medicines to treat diseases in humans and other animals. For example, a molecule called lovastatin—which is produced by a fungus known as *Aspergillus terreus*—can reduce a human patient's risk of heart disease. However, it is not known what role many secondary metabolites play in the microbe that produced them.

*A. terreus* lives in the soil, but it can also infect plants and animals. In addition to lovastatin, it also makes another secondary metabolite called terrein. A recent study identified the genes responsible for making terrein, and discovered that this molecule is harmful to plant cells and may help the fungus to colonize and thrive in the area immediately around plant roots, which is known as the rhizosphere.

Here, Gressler et al. studied how terrein may help the fungus to cope with competitors in this environment. The experiments show that terrein increases the availability of iron and inhibits the growth of competing microbes. A shortage of iron or nitrogen-containing nutrients can stimulate the fungus to produce terrein, and elevated levels of a molecule called methionine have the same effect. These conditions are commonly found in the rhizosphere and further experiments identified several proteins in the fungus that are required for sensing them.

Gressler et al.'s findings suggest that terrein helps to ensure that the fungus has sufficient nitrogen and iron to thrive in the rhizosphere. Also, this study confirms that the production of secondary metabolites in microbes can happen in response to elaborate cues from the environment, which may explain why only a limited number of secondary metabolites are produced by microbes when they are grown in the laboratory. Future studies will analyze other ways to activate the production of secondary metabolites outside of the microbe's normal environment, which may lead to the discovery of new important drugs.

but it is also a causative agent of life-threatening invasive aspergillosis in immunocompromised patients (*Baddley et al., 2003*; *Slesiona et al., 2012b*) and has recently been described as a pathogen of potato leaves (*Louis et al., 2013*).

While searching for a candidate protein required for pigment synthesis in *A. terreus* conidia, we serendipitously identified the gene cluster responsible for terrein production (*Zaehle et al., 2014*), which is the major SM formed by *A. terreus*. Terrein production is highly pronounced on sugar-rich plant-derived media such as potato dextrose broth (PDB) (*Zaehle et al., 2014*) and shows phytotoxic activities such as inhibition of seed germination and lesion formation on fruit surfaces (*Kamata et al., 1983*; *Zaehle et al., 2014*).

The terrein biosynthesis gene cluster consists of 11 genes (*terA–J*, *terR*), whereby *terA* encodes the key enzyme, which is a non-reducing polyketide synthase (*Zaehle et al., 2014*), and *terR* codes for the transcriptional activator TerR containing a GAL4-type $Zn_2Cys_6$ binuclear cluster DNA-binding domain (*Gressler et al., 2015a*), as frequently found in transcriptional activators of fungal SM gene clusters. Although activation of cluster genes is strictly TerR-dependent, no signals that lead to TerR activation under in vivo conditions have yet been identified.

Since terrein is the major SM produced by *A. terreus*—gram scales are easily achieved—we assumed a benefit from its production in the natural habitat. To elucidate this question, a detailed knowledge on the inducing factors stimulating terrein production and analyses of its biological activities were required. Therefore, we aimed to investigate environmental signals that result in *terR* expression and, eventually, the production of terrein. From these analyses, the impact of different global transcription factors on cluster induction was deduced. Two global transcription factors sense the quality and availability of nitrogen sources and specifically respond to the plant and fruit environment. Additionally, the iron responsive regulon plays a vital role in cluster induction, which indicates a specific contribution of terrein in modulating iron homeostasis.

## Results

### Fruits induce the production of terrein

In previous studies we showed that terrein is produced on plant-derived media such as PDB, which is in agreement with its phytotoxic biological activity (*Zaehle et al., 2014*). To address the question of specific conditions that induce the gene cluster, an *A. terreus* reporter strain was generated expressing the β-galactosidase gene *lacZ* from *Escherichia coli* under control of the terrein synthase promoter P*terA* (P*terA:lacZ*). Due to the dependence of P*terA* expression on TerR, this strain served as a direct indicator of *terR* expression and terrein production.

In agreement with a lack of terrein production, β-galactosidase activity was near the detection limit when the *A. terreus* P*terA:lacZ* strain was grown on glucose minimal medium. In contrast, and in agreement with previous observations, a 200–500 fold induction was observed on PDB medium (*Figure 1A*). Similarly, on Sabouraud and yeast extract-peptone-dextrose (YPD) medium, induction levels reached 10–30% compared with PDB. However, potato broth or casamino acids did not induce the cluster without the addition of glucose, indicating that glucose appears to be required for terrein production rather than repressing gene cluster induction as shown for other SM gene clusters (*Theilgaard et al., 1997*; *Brakhage et al., 2004*; *Gressler et al., 2011*). Indeed, when glucose medium was supplemented with 1% casamino acids as the nitrogen source, a 20–30 fold activation was detected. Since terrein can cause lesions on fruit surfaces and inhibits plant seed germination (*Kamata et al., 1983*; *Zaehle et al., 2014*), we assumed that sugar-rich fruit and root juices might have a strong stimulatory capacity. Therefore, we cultivated the reporter strain on banana, carrot, peach, and apple juice. β-Galactosidase activities from these media exceeded the activity of the already strong inducing PDB medium (*Figure 1B*) by a factor of at least five. Additionally, in subsequent LC analyses of culture extracts, a distinct ultraviolet signal for terrein was detected (*Figure 1—figure supplement 1A–D*). This led us to infect fresh bananas with the *A. terreus* Δ*akuB* strain (the parental strain for gene deletions; *Gressler et al., 2011*), a Δ*terR* mutant lacking the transcriptional activator, and a Δ*terA* deletion mutant that lacks the key polyketide synthase from the cluster.

Ethyl acetate extraction of bananas infected with Δ*akuB* strain revealed high amounts of terrein, whereas no terrein was detected after infection with the Δ*terR* or Δ*terA* mutant (*Figure 1C*). Similar results were obtained when fresh apples or nectarines were infected with the three *A. terreus* strains (*Figure 1—figure supplement 1E,F*). Furthermore, the extracts of bananas infected with *A. terreus* wild-type (and Δ*akuB*) caused strong lesion formation on fresh banana peels, which was hardly observed with extracts of bananas infected with the mutants (*Figure 1D*). Taken together, these results confirm terrein production in a natural habitat of fruit infection, although the specific factors leading to terrein production still remain unclear.

### Methionine supplementation acts as an inducing signal in non-inducing media

Since casamino acid-supplemented glucose medium revealed moderate gene cluster activation (*Figure 1A*), we assumed that specific amino acids could act as inducers. Therefore, we tested the inducing effect of amino acids as sole nitrogen sources by preparing six distinct pools that covered all 20 canonical proteinogenic amino acids (*Figure 2—figure supplement 1*).

While most of the pools did not activate *terA* expression, the pool with aliphatic amino acids caused moderate induction and media containing the sulphur-rich amino acids cysteine and methionine strongly induced gene expression, especially after 72 hr of cultivation (*Figure 2—figure supplement 1*). In the presence of the preferred nitrogen source glutamine, the induction by aliphatic amino acids was mainly lost. Also, the activation by cysteine was low and strongly concentration-dependent (*Figure 2A*). In contrast, methionine provoked a significant induction even at low concentrations. Therefore, we also analysed the effect of other sulphur-containing compounds in two concentrations (*Figure 2B*). However, no induction was observed when homocysteine, cystathionine, glutathione, dimethylsulfoxide, sulfate, or sulfide were tested. This indicates that methionine itself rather than a sulphur source causes the induction.

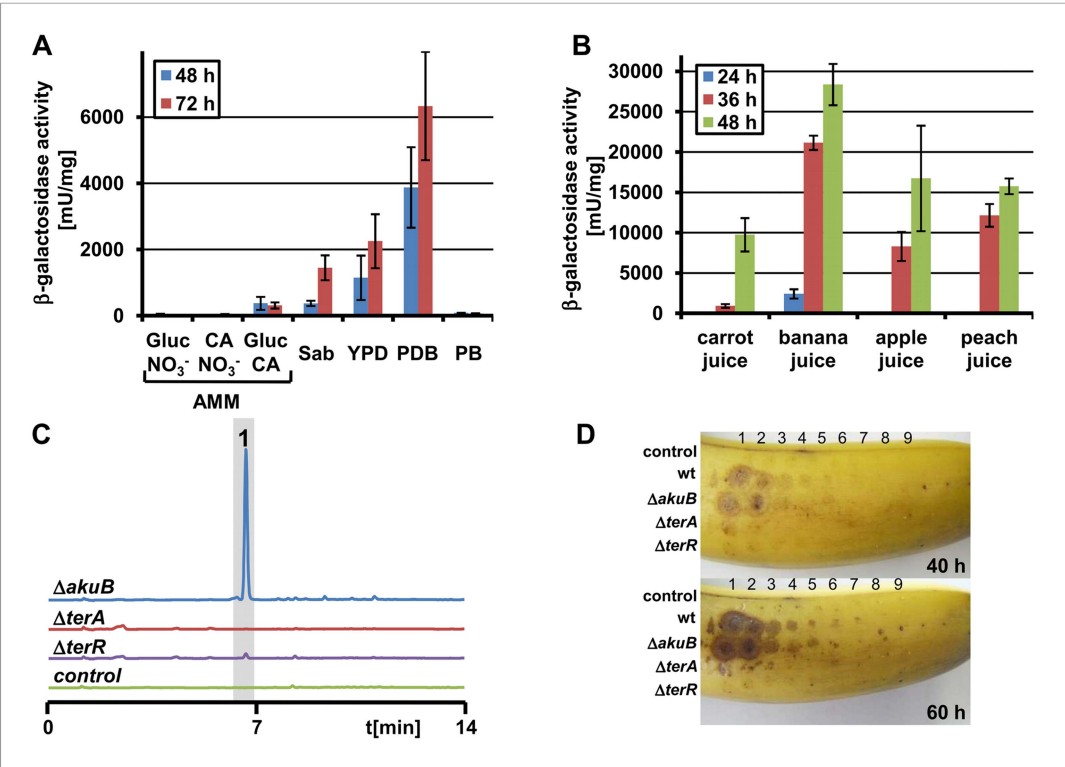

**Figure 1**. Terrein production and expression of *terA* on plant-derived media. (**A**) Promoter activity of strain P*terA:lacZ* after 48 hr and 72 hr on minimal media supplemented with glucose (Gluc), casamino acids (CA), or glucose + casamino acids (Gluc CA) or the complex media Sabouraud (Sab), yeast extract-peptone-dextrose (YPD), potato dextrose broth (PDB) or potato extract (PB). (**B**) Promoter activity of strain P*terA:lacZ* grown for 24, 36 and 48 hr in carrot, banana, apple, and peach juice. (**C**) High performance liquid chromatography analysis of banana extracts infected with *Aspergillus terreus* SBUG844 strains ΔakuB, ΔakuBΔterA or ΔakuBΔterR. A mock-infected fruit served as control. **1**—terrein. (**D**) Lesion formation on banana surfaces caused by extracts shown in (**C**). Photographs were taken after 40 and 60 hr. Lesions only occur with extracts from the wild-type and the parental strain of the mutants (ΔakuB). Numbers indicate the serial twofold dilution of the extracts starting from undiluted crude extracts down to 1:256 dilutions.

The following figure supplements are available for figure 1:

**Figure supplement 1**. High performance liquid chromatography (HPLC) analysis of extracts from *Aspergillus terreus* strains after cultivation in fruit juices and from infected nectarines and apples.

**Figure supplement 2**. β-Galactosidase activity of P*terA:lacZ* grown in banana juice without (control) or with different supplementations.

## Nitrogen starvation acts as a second inducing signal and promotes terrein production during fruit infection

Methionine and PDB medium induced the terrein biosynthesis gene cluster to a similar extent, but remained approximately five times below that of banana juice. This suggested that additional or alternative inducing signals might exist. Fruit juices are rich in sugars and the C:N ratio in fruits is very high, which could result in severe nitrogen limitation at a later growth state. Therefore, another set of experiments was performed in which the total concentration of amino acids was either high (50 mM) or low (10 mM) and P*terA* induction from the *lacZ* reporter strain was determined at different time points (*Figure 3A*). While high concentrations of amino acids or inorganic nitrogen sources such as nitrate or ammonia did not induce the cluster, low nitrogen contents resulted in a time-dependent 40–400 fold induction. To correlate gene expression with nitrogen exhaustion, the nitrogen consumption from a medium supplemented with 10 mM ammonium chloride was monitored and cluster induction was

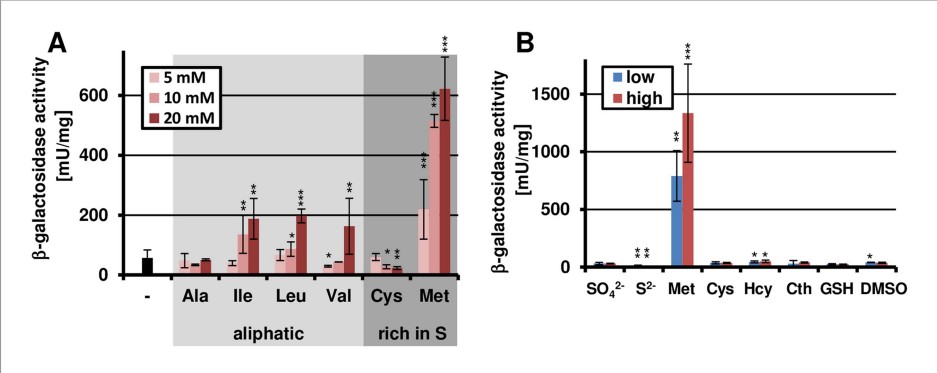

**Figure 2**. Methionine-dependent *terA* expression. (**A**) β-Galactosidase activity of SBUG844_P*terA:lacZ* after 48 hr of cultivation in glutamine-containing minimal media in combination with 5, 10, or 20 mM of the aliphatic (Ala, Ile, Leu, Val) or sulphur-containing amino acid (Cys, Met). Significance calculated against the glutamine control. (**B**) β-Galactosidase assay of SBUG844_P*terA:lacZ* in the presence of various sulphur sources. Glutamines containing minimal media were supplemented with low (5 mM) or high (10 mM) concentrations of $Na_2SO_4$, $Na_2S$, methionine (Met), cysteine (Cys), homocysteine (Hcy), reduced gluthathione (GSH), and dimethylsulfoxide (DMSO). Cystathionine (Cth) was used in final concentrations of 1 and 3 mM. Activity was determined after 48 hr of growth. Significance calculated against medium supplemented with sodium sulfate. All cultivations were performed in biological triplicates and activity determinations were made in technical duplicates. Statistical significance was calculated by the Student's paired t-test with a two-tailed distribution: *$p < 0.05$, **$p < 0.01$; ***$p < 0.001$.

The following figure supplement is available for figure 2:

**Figure supplement 1**. Amino acid-dependent *terA* expression.

simultaneously analysed. Indeed, as soon as nitrogen levels reached the detection limit, cluster expression was strongly induced (*Figure 3B*). Subsequently, a culture shift experiment was performed in which mycelium was pre-grown in the presence of 70 mM ammonium chloride and shifted to medium either with or without a nitrogen source (*Figure 3C*). While no reporter activity and terrein formation was observed from cultures of the nitrogen-rich medium, high β-galactosidase activity accompanied by terrein accumulation was observed in the nitrogen starvation medium.

These results confirmed that nitrogen starvation acts as an inducing factor that might be responsible for the high terrein production levels on fruit juices. In agreement, when ammonium chloride was added to banana juice, β-galactosidase activity was strongly reduced (*Figure 1—figure supplement 2*). Similarly, the P*terA:lacZ* reporter strain displayed high β-galactosidase activity when bananas were directly infected (*Figure 3E*). When bananas were supplemented with ammonium chloride prior to infection with the wild-type, ethyl acetate extracts of the bananas revealed a 50% reduction in terrein content compared with bananas without nitrogen supplementation (*Figure 3F*). This finding clearly indicates that nitrogen limitation is a major inducer for terrein production under natural conditions. However, it should be mentioned that the presence of sugars was always required, since especially hexoses from mono- and disaccharides provoked strong expression under nitrogen-limited conditions (*Figure 3D*), which is in direct contrast to the production of dihydroisoflavipucine in *A. terreus* that belongs to the class of fruit and root rot toxins. This metabolite is only produced in the strict absence of sugars and requires preferred nitrogen sources such as glutamine or asparagine for induction (*Gressler et al., 2011*).

## AreA is the main global nitrogen regulator in *A. terreus*

Nitrogen starvation and methionine marked important signals for terrein cluster induction. To unveil the global regulators that could be involved in signal transduction, the *A. terreus* genome was analysed for the presence of transcription factors known to play a role in nitrogen sensing, cross pathway control of amino acid synthesis, and stress response. The global nitrogen regulator AreA (ATEG_07264) (*Arst and Cove, 1973*; *Hynes, 1975*; *Davis et al., 2005*), the cross pathway control

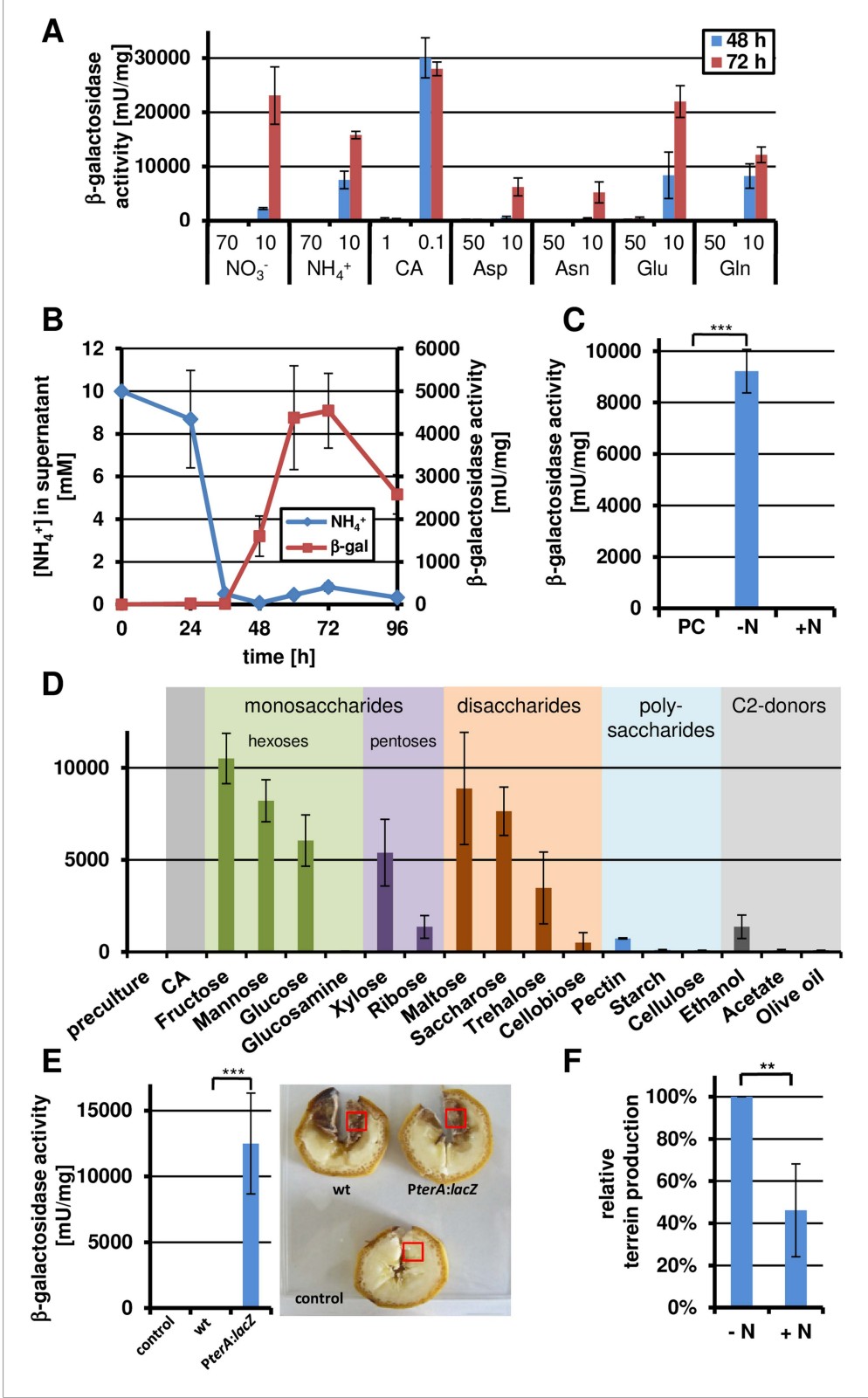

**Figure 3**. Terrein biosynthesis gene cluster activation under nitrogen starvation. (**A**) β-Galactosidase activity of SBUG844_P*terA:lacZ* was cultivated for 24, 36, 48, 72, and 96 hr in glucose minimal medium supplemented with different concentrations of various nitrogen sources: 70 mM and 10 mM NaNO₃ or NH₄Cl, 1% or 0.1% of casamino acids (CA), and 50 or 10 mM aspartate (Asp), asparagine (Asn), glutamate (Glu), or glutamine (Gln). (**B**) Correlation of
*Figure 3. continued on next page*

*Figure 3. Continued*

nitrogen consumption and *terA* promoter activity determined by β-galactosidase activity and ammonia consumption of SBUG844_P*terA:lacZ* in glucose minimal medium with 10 mM NH₄Cl. (**C**) β-Galactosidase activity of SBUG844_P*terA:lacZ* in nitrogen shift experiments. Cultures grown for 48 hr in glucose minimal medium with 70 mM NH₄Cl (PC) were washed and transferred to medium with (+N) or without 70 mM NH₄Cl (−N) and promoter activity was determined after 15 hr of cultivation. (**D**) Carbon source dependent *terA* promoter activation under nitrogen starved conditions. Strain SBUG844_P*terA:lacZ* was pre-cultured for 48 hr on casamino acids without sugar supplementation. The mycelium was washed and transferred to nitrogen-free media with different carbon sources. Reporter activity was determined 24 hr after the shift. (**E**) β-Galactosidase activity from bananas infected with conidia suspension of SBUG844 wild-type and P*terA:lacZ*. Sections (red boxes) were cut from bananas, ground to a fine powder and subjected to β-galactosidase activity determination. Activity was only detected from the reporter strain. (**F**) Quantification of terrein from wild-type infected bananas with or without ammonium supplementation. All tests were performed in biological triplicates and technical duplicates; p values were calculated by the Student's paired t-test with a two-tailed distribution: **p<0.01.

regulator CpcA (ATEG_03131) (*Hoffmann et al., 2001*; *Krappmann et al., 2004*), the stress response bZIP transcription factor AtfA (ATEG_04664) (*Balazs et al., 2010*; *Lara-Rojas et al., 2011*), and the nitrogen starvation-induced ras-protein RhbA (ATEG_09480) (*Panepinto et al., 2002*) were selected for gene deletions. All mutants were tested for their growth properties in the presence of proteinogenic amino acids or ornithine, citrulline, urea, and casamino acids, all of which were used as the sole nitrogen source. Additionally, several complex media were analysed (*Figure 4—figure supplement 1*).

Only minor growth defects were observed with the *cpcA* and *rhbA* mutants, with some general growth reduction on selected amino acids (Asp, His, Ser, Thr) or complex media. However, among all the mutants tested, the *areA* mutant revealed the most severe growth defects. This mutant grew like the wild-type on glutamine and displayed some reduced growth on the nitrogen-rich amino acids asparagine, aspartate, and histidine as well as on urea and ammonium chloride. However, the Δ*areA* strain was unable to use any other proteinogenic amino acid. As described for other *Aspergillus* species (*Hunter et al., 2014*), these results confirm an essential role of AreA in nitrogen sensing and utilisation in *A. terreus*. In contrast, the *atfA* mutant was only impaired in growth at high aspartate concentrations (50 mM). However, conidia of this mutant lacked the typical yellow-brown pigmentation. Additionally, the *atfA* mutant completely lacked the typical red colouration of the medium in the presence of methionine, which has previously been demonstrated to be associated with terrein production (*Zaehle et al., 2014*). This implied a regulatory role of *atfA* on SM production in *A. terreus*. On the contrary, the *atfA* mutant did not show increased sensitivity against oxidative or osmotic stress as described for other *Aspergillus* species (*Balazs et al., 2010*; *Lara-Rojas et al., 2011*). Finally, a double deletion of the *areA* and *atfA* genes was generated (Δ*areA*Δ*atfA*) resembling the growth phenotypes of both single mutants, since it was only able to grow on the media that supported growth of the *areA* mutant where it formed white conidia, as observed for the Δ*atfA* strain.

## The global transcription factors AreA and AtfA are essential for terrein biosynthesis gene cluster induction during nitrogen starvation

To test the effect of transcription factor mutations on terrein biosynthesis gene cluster activation, all mutants were pre-grown on non-inducing glucose medium with glutamine and transferred to medium with or without nitrogen. Terrein was quantified after 24 hr (*Figure 4A*). The wild-type, the Δ*cpcA*, and the Δ*rhbA* mutant revealed low terrein production in the presence of nitrogen, but high terrein titers when nitrogen was omitted. This indicates that *cpcA* and *rhbA* are dispensable for terrein production. In contrast, both the Δ*areA* and the Δ*atfA* mutant only produced marginal amounts of terrein under nitrogen-limited conditions (*Figure 4A*), and this effect was completely cured in the complemented mutants *areA*[c] and *atfA*[c]. No terrein was detected in a Δ*areA*Δ*atfA* double mutant. Therefore, both *areA* and *atfA* appear essential for terrein production under nitrogen limitation. To confirm this assumption, we expressed the cluster-specific transcription factor gene *terR* under control of the *gpdA* promoter in a Δ*areA*Δ*atfA* mutant background. A constitutive terrein production was observed, indicating that AreA and AtfA regulate *terR* expression, but are not essential for expression of the structural genes which directly depend on TerR (*Figure 4A*). Second, we analysed the effect of *areA* and *atfA* deletion on gene

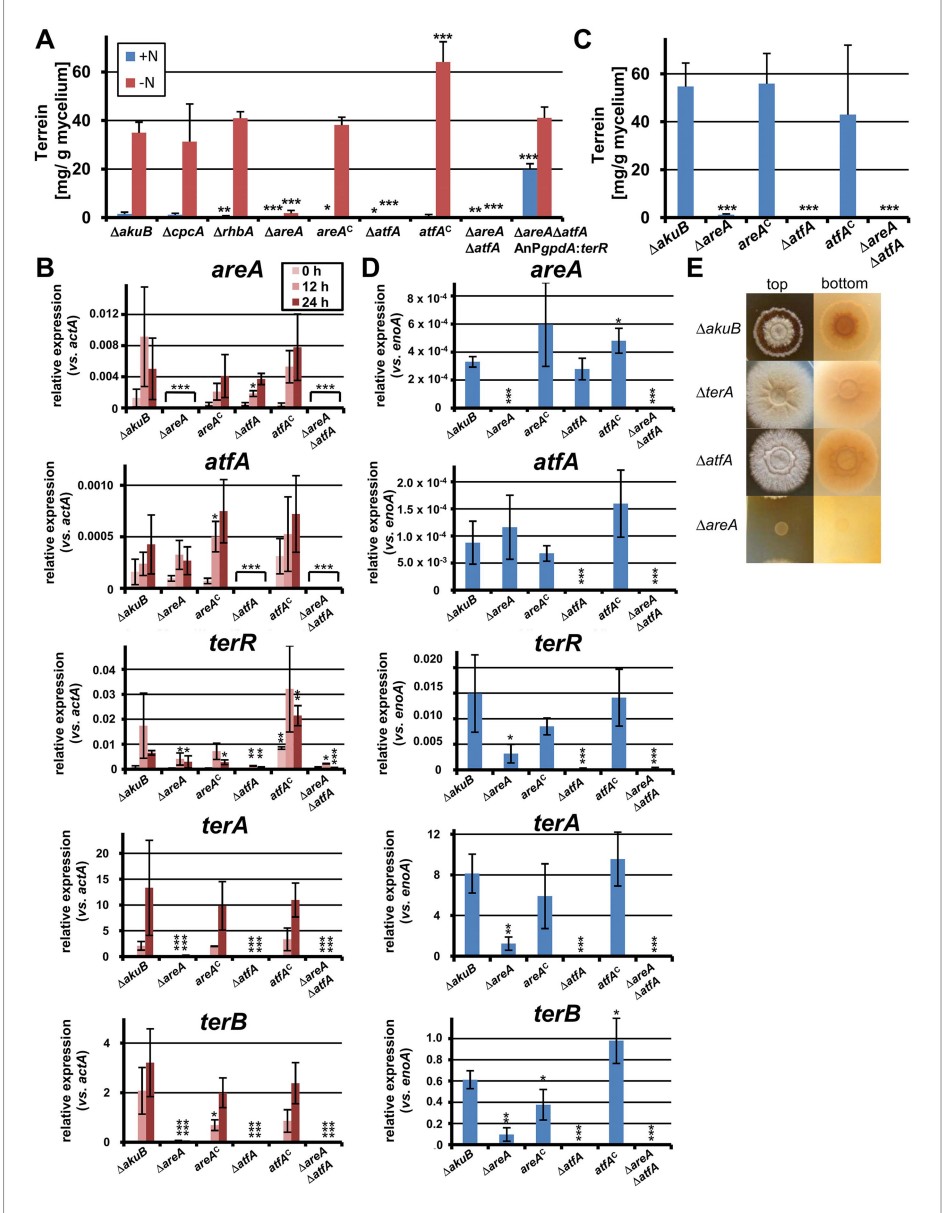

**Figure 4**. Terrein quantification and expression of terrein cluster genes in nitrogen regulator mutants. (**A**) Terrein quantification from the parental strains SBUG844ΔakuB (ΔakuB), regulator mutants (ΔcpcA), (ΔrhbA), (ΔareA), (ΔatfA), (ΔareAΔatfA), complemented mutants (areA^C), (atfA^C), and strain SBUG844ΔakuBΔareAΔatfAΔ::AnPgpdA:terR with terR overexpression in the ΔareAΔatfA background. Mycelia were pre-grown in glutamine-supplemented media, washed and transferred to minimal medium with (+N) or without (−N) 50 mM glutamine. Terrein was quantified from supernatants 24 hr after the shift. (**B**) qRT-PCR of strains ΔakuB, ΔareA, areA^C, ΔatfA, atfA^C, and ΔareAΔatfA were pre-cultivated for 40 hr in glutamine-supplemented media and transferred to nitrogen starvation. RNA was isolated at 0, 12, and 24 hr of starvation. Transcript levels were normalised against the actin gene *actA* by fold expression = $2(C_T^{target} − C_T^{actA})$.
(**C**) Terrein quantification from strains shown in (**B**) after 72 hr of cultivation in glutamine-containing minimal medium supplemented with 10 mM methionine. (**D**) qRT-PCR from RNA of strains shown in (**C**) isolated after 48 hr of cultivation. Transcript levels were normalized against the enolase gene *enoA* by fold expression = $2(C_T^{target} − C_T^{enoA})$. (**E**) Top and bottom view of colonies of *Aspergillus terreus* wild-type (ΔakuB) and mutants (ΔterA, ΔatfA, ΔareA) grown for 72 hr on solid minimal media supplemented with 25 mM methionine as sole nitrogen source. The red pigmentation of the wild-type (bottom view) is lost in the ΔterA and ΔatfA mutants that show some enhanced growth while unable to produce terrein. The ΔareA strain is unable to grow. In all experiments biological triplicates with technical duplicates were analysed. Statistical significances in comparison to the parental ΔakuB strain were calculated by the Student's paired t-test with a two-tailed distribution: *p<0.05; **p<0.01; ***p<0.001.

*Figure 4. continued on next page*

*Figure 4. Continued*

The following source data and figure supplements are available for figure 4:

**Source data 1**. Genotypes of strains used in the study.

**Source data 2**. List of oligo nucleotides used in the study.

**Figure supplement 1**. Analysis of colony formation and growth phenotypes of nitrogen regulator mutants in the presence of different nitrogen sources

**Figure supplement 2**. Biochemical characterisation of the recombinant AreA DNA-binding domain and in vitro binding to HGATAR motifs identified in the *terR* promoter of the *Aspergillus terreus* terrein biosynthesis gene cluster.

**Figure supplement 3**. High performance liquid chromatography analyses from culture filtrates of SBUG844 wild-type and two independent *atfA* overexpression mutants (AnP*gpdA*:*atfA*; OE 1 and 2).

**Figure supplement 4**. Southern blot analyses for *Aspergillus terreus* strains generated in this study.

**Figure supplement 5**. Terrein production and susceptibility to osmotic stress in a *sakA* mutant.

---

expression by qRT-PCR from strains shifted for 0 hr, 12 hr, and 24 hr to nitrogen starvation. qRT-PCR was performed on the cluster genes *terA*, *terB*, the specific activator *terR*, and the global transcription factors *areA* and *atfA* (*Figure 4B*).

In the wild-type the regulators *areA* and *atfA* showed a time-dependent increase in gene expression. While the regulator *terR* was most strongly upregulated after 12 hr, expression of the TerR-controlled terrein biosynthesis genes *terA* and *terB* continued to increase after 12 hr and reached 13.3 and 3.2 times the expression level of the actin control gene at 24 hr (*Figure 4B*). On the contrary, while complemented mutants behaved like the wild-type, the *areA* and *atfA* deletions strongly reduced activation of *terR* and, in turn, the expression of *terA* and *terB*. Results from qRT-PCR perfectly coincided with the substantially reduced terrein production rates under nitrogen limitation in these mutants (*Figure 4A*). AreA recognises the DNA-binding motif HGATAR, and two adjacent binding sites are generally required for transcriptional activation due to dimer formation of AreA monomers (*Ravagnani et al., 1997*). In this respect, the *terR* promoter contains two putative AreA binding sites that match the HGATAR consensus (BS1 and BS2; positions −59 and −72 relative to the translational start point). Surface plasmon resonance (SPR) analyses with the *Aspergillus nidulans* AreA DNA-binding domain, which is 91% identical to the respective *A. terreus* AreA domain, showed that BS1 and BS2 are recognised with high affinity (*Figure 4—figure supplement 2*). This strengthens the model of a direct involvement of AreA in the activation of *terR* expression. However, the reduction of *terR* expression was less pronounced in the Δ*areA* than in the Δ*atfA* background (*Figure 4B*), which indicates that AreA is not the only activator acting on the *terR* promoter. Therefore, we additionally searched for putative palindromic AtfA/Sko1 binding sites (5′-TKACGTMA-3′) in the promoter regions of the cluster (*Proft et al., 2005*). Only one hit (TGACGTCA) was identified in the promoter of the structural gene *terC*. However, if one mismatch is allowed, there is a putative binding site at position −731 relative to the ATG start codon of *terR* (5′-TGGCGTCA-3′), but it remains speculative whether this binding site is recognised by *A. terreus* AtfA. Nevertheless, it should be mentioned that even a single half site of the suggested motif could promote transcription factor binding and promoter induction (*Proft et al., 2005*). We therefore conclude that, although direct evidence for AtfA binding at the *terR* promoter is lacking, both transcription factors seem to regulate *terR* expression. In agreement, *terR*, *terA*, and *terB* expression and terrein production showed the strongest decrease in the Δ*areA*Δ*atfA* double knock-out mutant (*Figure 4A*). In addition, a constitutive expression of *atfA* by the *gpdA* promoter led to increased terrein production (*Figure 4—figure supplement 3*). This supports our hypothesis of direct involvement of AtfA in terrein cluster regulation.

## AreA and AtfA mediate terrein biosynthesis gene cluster induction in the presence of methionine

Due to the requirement for *atfA* and *areA* in terrein biosynthesis gene cluster induction under nitrogen starvation, we also tested their contribution to methionine-dependent induction in nitrogen-supplemented medium (*Figure 4C*). While the wild-type strain and complemented mutants showed high terrein production rates of up to 55 mg/g mycelium, the terrein levels in the Δ*areA*, Δ*atfA*, or Δ*areA*Δ*atfA* double mutant remained near the detection limit and qRT-PCR was performed to confirm this result on the transcriptional level.

Although *areA* and *atfA* were expressed in the wild-type only at low levels on glutamine/methionine medium (*Figure 4D*), deletion of *areA* or *atfA* reduced *terR* and, consequentially, *terA* and *terB* expression. In the Δ*atfA* mutant *terR* transcription was completely abolished and, in agreement, transcription of *terA* and *terB* was no longer detected. An *areA* mutant is unable to use methionine as a nitrogen source (*Figure 4E*), and its uptake may be limited leading to the loss of transcriptional activation. In contrast, the *atfA* mutant still uses methionine as a nitrogen source, but neither produces terrein nor pigmented conidia nor the red colouration of the medium which is associated with terrein production in the presence of methionine (*Figure 4E*). This implies that *atfA* may be induced by a methionine-dependent signaling cascade that subsequently leads to *terR* expression.

## Iron limitation acts as a third independent signal for terrein cluster induction

While nitrogen starvation stimulated terrein production, starvation of other macroelements such as carbon, sulphur, or phosphate did not result in terrein production (data not shown). However, this did not exclude limitation of trace elements as inducing signals. Therefore, the β-galactosidase activity from the P*terA:lacZ* reporter strain was determined from cultures with reduced amounts of trace elements. Indeed, a decrease in trace elements was accompanied by increased *terA* promoter activity (*Figure 5A*). To attribute this activation to a specific trace element, media with limited amounts of trace elements were supplemented with each of the single trace elements $FeSO_4$, $ZnSO_4$, $CuSO_4$, $MnCl_2$, $Na_2MoO_4$, $CoCl_2$, and $H_3BO_3$. Cluster induction was observed in all cultures except for that supplemented with $FeSO_4$ (*Figure 5B*), indicating cluster activation from iron limitation. Subsequently, a minimal medium was prepared that contained all trace elements but iron which led to strong cluster induction, and the addition of 40 μM $FeCl_3$ completely repressed the induction (*Figure 5C*). Thus, besides methionine and the limitation of nitrogen, a lack of iron—but no other trace element—induces the terrein biosynthesis gene cluster.

## Iron starvation induces iron assimilation pathways

Fungal iron acquisition has been well investigated in *A. nidulans* (*Eisendle et al., 2003*) and *Aspergillus fumigatus* (*Haas, 2014*), but only limited information was available on the iron acquisition systems from *A. terreus*. A BLASTp search for orthologous genes in the genome of *A. terreus* NIH2624 (*Table 1*) revealed that all genes required for a reductive iron assimilation (RIA) system (*fetC*, *freB*, *ftrA*), siderophore biosynthesis (*sid* genes, except *sidG*), siderophore transport (*mirA-D*, *sitA*), and regulators of iron homeostasis (*srbA*, *sreA*, and *hapX*) are well conserved in the *A. terreus* genome. When tested under iron limitation, qRT-PCR analyses confirmed a strong induction of genes from siderophore biosynthesis, siderophore transport, and the RIA pathway (*Figure 5—figure supplement 1*). To confirm siderophore production by *A. terreus*, we cultivated the wild-type strain under iron-rich (200 μM $FeCl_3$) and iron-limited conditions (no or 20 μM $FeCl_3$) and identified ferrichrysin (**2**) and coprogen (**3**) only under iron limitation (*Figure 5D* and *Figure 5—source data 1, 2*) (*Zähner et al., 1963*; *Bertrand et al., 2010*). Coprogen revealed a higher stability than ferrichrysin. Subsequent analyses involving siderophores were therefore based only on coprogen. When terrein was compared with purified coprogen for chelating iron, only coprogen showed the expected iron chelating activity in CAS agar plate assays (*Figure 5—figure supplement 2*). Thus, although terrein is produced under iron limitation, it does not depict a new kind of siderophore.

## Terrein reduces ferric to ferrous iron by its oxidation to propylene maleic acid

Since some antioxidative properties of terrein had previously been described (*Trabolsy et al., 2014*), the ability of terrein in reducing ferric ($Fe^{3+}$) to ferrous ($Fe^{2+}$) iron was investigated by using the ferrous

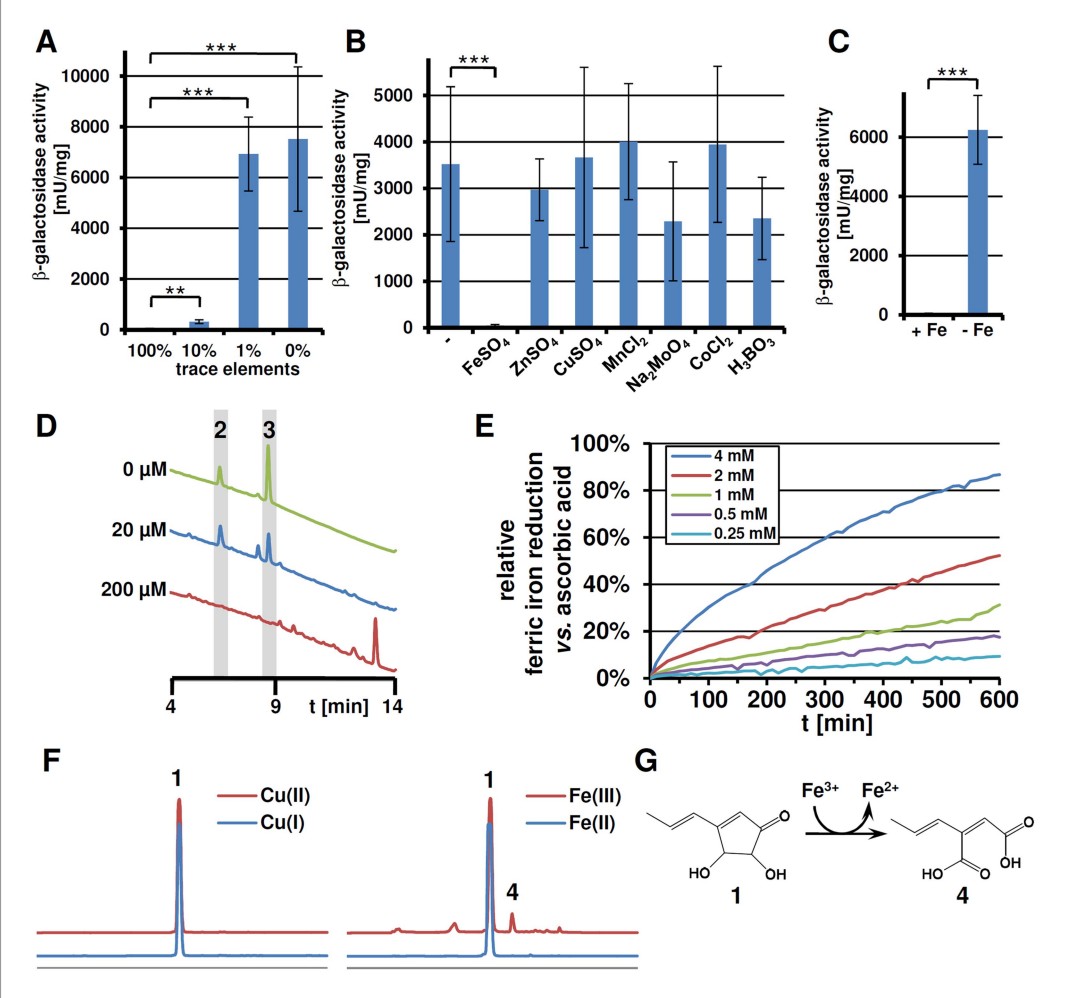

**Figure 5**. Expression of *terA*, production of siderophores under iron limited conditions and iron reducing properties of terrein. All analyses were performed in biological triplicates and technical duplicates. Statistical significances were calculated by the Student's paired t-test with a two-tailed distribution. (**A**) β-Galactosidase activity from SBUG844_P*terA:lacZ* in AMM(-N)G100Gln50 medium with limited amounts of trace elements. Asterisks indicate p values vs 100% trace elements: **p<0.01; ***p<0.001. (**B**) β -Galactosidase activity from SBUG844_P*terA:lacZ* in AMM(-N)G100Gln50 medium with 1% trace elements supplemented with 20 μM of specific trace elements: FeSO₄, ZnSO₄, CuSO₄, MnCl₂, Na₂MoO₄, CoCl₂, or H₃BO₃. Asterisks indicate p values vs activity without supplementation: ***p<0.001. (**C**) β-Galactosidase activity from SBUG844_P*terA:lacZ* in AMM(-N)G100Gln50 medium with and without 40 μM FeCl₃. Asterisks indicate p values vs 40 μM FeCl₃: ***p<0.001. (**D**) High performance liquid chromatography (HPLC) profiles of lyophilised culture supernatants of Δ*terA* after 72 hr of cultivation in AMM(-N)G100Gln50 supplemented with 0, 20, or 200 μM FeCl₃. Peaks for the siderophores ferrichrysin (**2**) and coprogen (**3**) are indicated. (**E**) Fe(III) reduction by terrein determined by the TPTZ assay. Ascorbic acid served as control and maximum reduction by ascorbic acid was set as 100%. Analyses were made from duplicates. (**F**) HPLC profiles of terrein after incubation with different oxidised (upper, red lines) or reduced (lower, blue lines) metal ions. Left: Cu(II)SO₄ and Cu(I) Ac; right: Fe(III)Cl₃ and Fe(II)SO₄. **1** = terrein; **4** = propylene maleic acid. (**G**) Scheme of terrein oxidation during iron reduction leading to the formation of propylene maleic acid.

The following source data and figure supplements are available for figure 5:

**Source data 1**. Analytical data of coprogen.

**Source data 2**. Analytical data of ferrichrysin.

**Figure supplement 1**. qRT-PCR expression analysis of genes from iron acquisition systems under iron-supplemented and limited conditions.

*Figure 5. continued on next page*

*Figure 5. Continued*

**Figure supplement 2**. Chrome azol S (CAS) assay of coprogen and terrein.

**Figure supplement 3**. pH- and time-dependent Fe(III) reduction by terrein assayed by TPTZ.

**Figure supplement 4**. $^1$H NMR (500 MHz, MeOD; upper panel) and $^{13}$C NMR (150 MHz, MeOD; lower panel) of compound 4, 2-((E)-prop-1-en-1-yl)maleic acid.

**Figure supplement 5**. HSQC NMR (600 MHz, MeOD; upper panel) and HMBC NMR (600 MHz, MeOD; lower panel) of compound 4, 2-((*E*)-prop-1-en-1-yl)maleic acid.

iron chelator 2,4,6-tripyridyl-S-triazine (TPTZ). The strong antioxidant ascorbic acid (*Elmagirbi et al., 2012*) served as control (*Figure 5E*). Although ascorbic acid showed a much higher reducing potential, terrein was also able to convert ferric to ferrous iron in a concentration-dependent (*Figure 5E*) and pH-dependent (*Figure 5—figure supplement 3*) manner with a pH optimum of 3–4. In this respect, we noticed a significant fall in the culture pH under iron limitation from 6.5 to about 3.5, which agrees with the optimum pH for terrein-mediated iron reduction. Thus, since terrein is produced during growth under iron limitation, the combination with a fall in pH could indeed increase iron availability.

To elucidate the structure of the terrein oxidation product, the terrein-mediated iron reduction assay was scaled up and extractions were subjected to high performance liquid chromatography (HPLC) analysis. Here, besides a major proportion of terrein that remained in its original structure (*Figure 5F*), a new peak (**4**) was observed, which was identified by NMR analyses (*Figure 5—figure supplements 4, 5*) as 2-((*E*)-prop-1-en-1-yl) maleic acid (PMA, *Figure 5G*). The structure of the oxidation product implies that oxidation took place at the two hydroxylated carbon atoms of terrein, leading to a cleavage of the pentenone ring system as proposed by *Trabolsy et al. (2014)*. In contrast, conversion of terrein to PMA was not observed during incubation with $Fe^{2+}$, confirming that iron reduction was the cause of PMA formation. No conversion of terrein was observed with oxidised copper ions ($Cu^{2+}$), indicating a limited reductive potential which is sufficient for iron but not for copper (*Figure 5F*).

In conclusion, while terrein cannot chelate iron to support the siderophore-mediated iron uptake, its reductive potential is sufficient to reduce ferric to ferrous iron, which may increase iron solubility and could ease the direct uptake of iron via the ferrous iron transport system.

## The iron response regulator HapX regulates siderophore and terrein biosynthesis in *A. terreus*

Due to the co-regulation with siderophore biosynthesis, we assumed that transcriptional regulators involved in iron homeostasis could also regulate terrein biosynthesis. In *A. nidulans* and *A. fumigatus*, siderophore biosynthesis is regulated by HapX, a transcriptional inducer under iron limitation, and SreA, a repressor in the presence of iron (*Haas, 2012*).

To investigate the impact of SreA and HapX on siderophore (coprogen) and terrein synthesis, the *sreA* gene was completely and the *hapX* gene partially deleted (the latter due to incomplete sequence information at the *hapX* locus tag ATEG_08014). Additionally, while the *sreA* mutant was complemented with the *A. terreus sreA* gene, the *hapX* mutant was complemented with the *hapX* gene from *A. nidulans* FGSC A4. All strains were cultivated in iron-supplemented and iron-limited media and terrein and coprogen were quantified from culture supernatants (*Figure 6A,B*). All complemented mutants behaved similar to the wild-type with production of only marginal amounts of coprogen and terrein in the presence of iron. Under iron starvation the production rates for both metabolites strongly increased.

On the other hand, partial *hapX* deletion reduced coprogen production under iron limitation by approximately 50% and terrein concentrations by about 90% (*Figure 6A,B*). This indicates that HapX directly activates both siderophore and terrein biosynthesis in *A. terreus*. In contrast, the Δ*sreA* mutant produced significantly higher amounts of coprogen in the presence of iron whereas terrein production did not significantly increase (*Figure 6A,B*). This confirms SreA as a negative feedback regulator in siderophore biosynthesis, whereas it does not control terrein production. In contrast, HapX positively controls both pathways.

**Table 1**. BLASTp analysis of homologous genes for iron uptake and regulation of iron homeostasis (adapted by [**Haas, 2012**])

| *Aspergillus fumigatus* | function | Gene code‡ | Expression | *Aspergillus terreus* gene code | Identity/similarity |
|---|---|---|---|---|---|
| **Reductive iron assimilation (RIA )** | | | | | |
| FetC | Ferroxidase | AFUA_5G03790 | −Fe | **ATEG_08032** | 79%/89% |
| FreB | Ferric reductase | AFUA_1G17270 | −Fe | **ATEG_10322** | 53%/64% |
| FtrA | Iron permease | AFUA_5G03800 | −Fe | **ATEG_08031** | 75%/84% |
| **Siderophore biosynthesis (SB)** | | | | | |
| EstA | TAFC esterase | AFUA_3G03660 | −Fe | ATEG_04072 | 44%/58% |
| NpgA/PptA | Phosphopantetheinyl transferase | AFUA_2G08590 | − | ATEG_09695 | 56%/65% |
| SidA | Ornithine monooxygenase | AFUA_2G07680 | −Fe | **ATEG_06879** | 78%/85% |
| SidC | FC NRPS | AFUA_1G17200 | −Fe | **ATEG_05073** | 60%/76% |
| SidD | FSC NRPS | AFUA_3G03420 | −Fe | **ATEG_07488** | 43%/59% |
| SidF | Transacylase | AFUA_3G03400 | −Fe | ATEG_05075 | 52%/67% |
| SidG | Transacetylase | AFUA_3G03650 | −Fe | none | − |
| SidH | Mevalonyl hydratase | AFUA_3G03410 | −Fe | ATEG_01509 | 53%/67% |
| SidI | Mevalonyl ligase | AFUA_1G17190 | −Fe | **ATEG_05074** | 86%/91% |
| SidL | Transacetylase | AFUA_1G04450 | − | ATEG_03770 | 64%/76% |
| **Siderophore transporter (SIT)** | | | | | |
| MirA | Enterobactin transporter | AN7800; - | −Fe | **ATEG_04071** | 68%/77% |
| MirB | TAFC transporter | AN8540; -AFUA_3G03640 | −Fe | **ATEG_02711** | 50%/68% |
| MirC* | | AN7485; AFUA_2G05730 | −Fe | **ATEG_06762** | 78%/87% |
| MirD† | Trichotecene efflux pump | AFUA_3G03440 | − | **ATEG_07487** | 40%/58% |
| SitA/SitT* | | AN5378; AFUA_7G06060 | −Fe | **ATEG_06329** | 62%/73% |
| **Regulatory proteins** | | | | | |
| HapX | bZip-TF | AFUA_5G03920 | −Fe | **ATEG_08014** | 77%/83% (hapX re-annotated) |
| SreA | GATA TF | AFUA_5G11260 | +Fe | **ATEG_07741** | 67%/75% |
| SrbA† | HLH TF | AFUA_2G01260 | −Fe | ATEG_08156 | 72%/82% |

Genes selected for qPCR analyses are highlighted in bold.
*Genes annotated according to (**Schrettl et al., 2008**).
†Genes annotated according to (**Blatzer et al., 2011**).
‡Gene codes ANxxxx refer to the *A. nidulans* genome.

These results were confirmed on the transcriptional level by qRT-PCR (*Figure 6C* and *Figure 6—figure supplement 1*). The inactivation of *hapX* prohibited *terR* transcription and resulted in an inability to induce terrein cluster genes *terA*, *terB*, and *terC*. In contrast, deletion of *sreA* influenced neither *terR* activation nor expression of other terrein cluster genes in the presence of iron. We therefore conclude that *hapX* is the major regulator for terrein cluster induction under iron starvation.

To elucidate the complete sequence of the *hapX* gene we subsequently used degenerate primers to amplify the main proportion of *hapX* from cDNA. The complete sequence of the *A. terreus hapX* gene is found under accession number KP233834 (*Gressler et al., 2015b*). The sequence of the full-length HapX protein matches with that of HapX proteins from other *Aspergillus* species such as *A. nidulans* (73% identity), *A. fumigatus* (77%), *Aspergillus niger* (81%), and *A. oryzae* (81%).

## Terrein supports growth in the absence of the siderophore system

In general, siderophore-based iron acquisition is highly efficient and assumed to be more important than the reductive iron assimilation pathway. In *A. fumigatus*, growth and virulence defects caused by

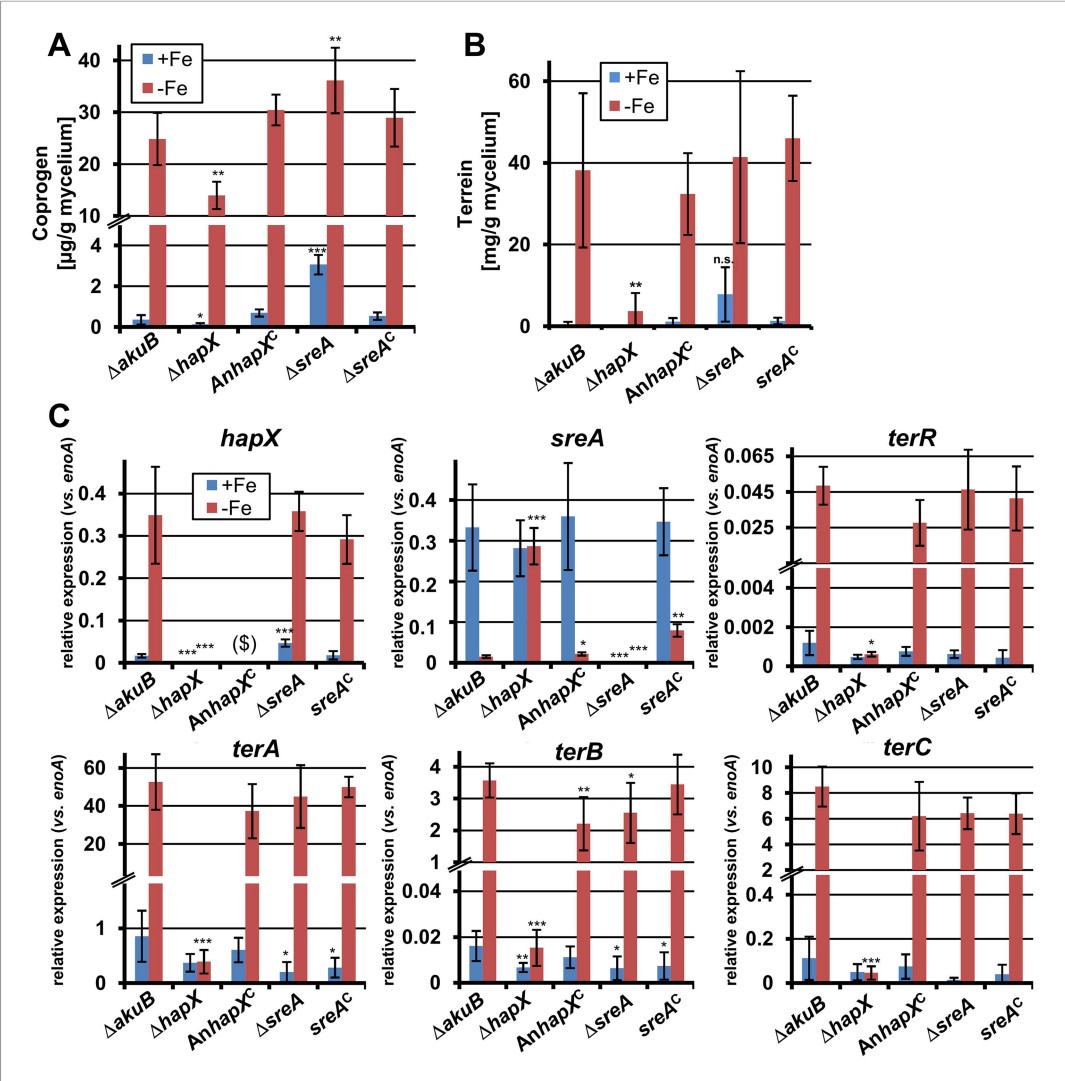

**Figure 6**. Effect of *hapX* and *sreA* deletion on terrein and coprogen biosynthesis. (**A**) Coprogen or (**B**) terrein quantification from SBUG844Δ*akuB* (Δ*akuB*), SBUG844Δ*akuB*Δ*hapX* (Δ*hapX*), SBUG844Δ*akuB*Δ*hapX*/An*hapX*$^C$ (An*hapX*$^C$), SBUG844Δ*akuB*Δ*sreA* (Δ*sreA*), and SBUG844Δ*akuB*Δ*sreA*/*sreA*$^C$ (*sreA*$^C$) grown for 72 hr in AMM_G100Gln50 with (+Fe) or without 40 µM FeCl$_3$ (–Fe). Coprogen was quantified from lyophilised culture supernatants and terrein from culture extracts. (**C**) qRT-PCR from strains and media described in (**A**) and (**B**). RNA was isolated after 40 hr of cultivation. Transcript levels were normalised against *enoA* by fold expression = 2($C_T^{target} − C_T^{enoA}$). ($) denotes the lack of *hapX* transcripts from the complemented Δ*hapX* strain, since the *Aspergillus nidulans hapX* was used for complementation. qRT-PCR on the complemented mutant with oligonucleotides specific for An*hapX* are shown in *Figure 6—figure supplement 1*. All analyses were performed from biological triplicates and technical duplicates. Statistical significances were calculated in comparison to the parental Δ*akuB* strain by the Student's paired t-test with a two-tailed distribution: *p<0.05; **p<0.01; ***p<0.001.

The following figure supplement is available for figure 6:

**Figure supplement 1**. qRT-PCR analysis of *Aspergillus nidulans hapX* expression in the *Aspergillus terreus* wild-type SBUG844Δ*akuB*, the *hapX* mutant SBUG844Δ*akuB*Δ*hapX* and its complemented strain SBUG844Δ*akuB*Δ*hapX*/An*hapX*$^C$.

the interruption of the reductive iron assimilation pathway are only observed when the siderophore-based system is also inactivated (*Blatzer et al., 2011*). Therefore, to elucidate a positive effect of terrein on iron acquisition, we deleted the *sidA* gene in *A. terreus* that encodes the L-ornithine-$N^5$-monooxygenase, a key

enzyme in hydroxamate siderophore biosynthesis. Coprogen production was confirmed in the wild-type and a complemented mutant, but was completely lacking from the ΔsidA mutant (*Figure 7—figure supplement 1*). When analysed for growth phenotypes on solid media, all complemented mutants and the ΔterA strain behaved like the wild-type (*Figure 7A* and *Figure 7—figure supplement 1A*). The hapX

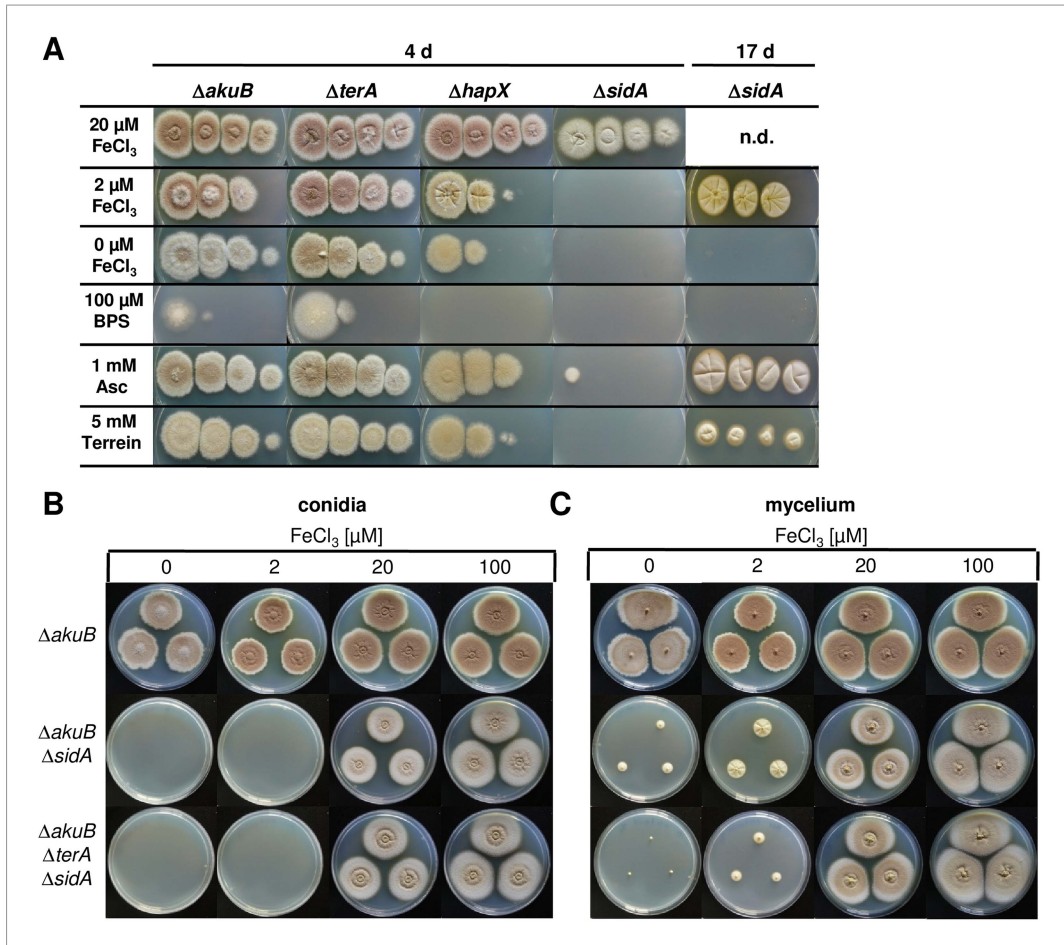

**Figure 7**. Siderophore production in *sidA* mutants and growth-supporting effect of terrein under iron limitation. (**A**) Growth of wild-type ΔakuB, ΔterA, ΔhapX, and ΔsidA on AMM(−N)G100Gln10 plates containing different iron concentrations. Photographs were taken after 4 d and, as indicated, after 17 d of incubation at 37°C. Iron-free medium was supplemented with 100, 20, 2, or 0 μM FeCl₃. Additionally, plates without iron were supplemented with bathophenanthroline disulfonate (BPS; 100 μM) to simulate harsh starvation conditions or with either ascorbic acid (1 mM) or terrein (5 or 10 mM). (**B**, **C**) Impact of terrein production on growth of ΔsidA mutants. AMM-G100Gln50 plates were supplemented with 0, 2, 20, or 100 μM FeCl₃ and either inoculated with (**B**) conidia or (**C**) mycelial pellets from cultures pre-grown for 40 hr in AMM-G100Gln50 with 200 μM FeCl₃. The parental *Aspergillus terreus* wild-type ΔakuB, the ΔakuBΔsidA strain, and the ΔakuBΔterAΔsidA are shown. Mycelia from a 40 hr culture in AMM-G100Gln50 with 200 μM FeCl₃ was washed with iron-free medium and three pellets were applied to the plates. Plates were incubated at 37°C for 5 d.

The following figure supplements are available for figure 7:

**Figure supplement 1**. Dependence of colony formation of *Aspergillus terreus* mutants on iron availabilty, ascorbic acid, terrein, and coprogen.

**Figure supplement 2**. Antifungal activity of terrein and potato dextrose broth (PDB) medium from *Aspergillus terreus* wild-type cultivations.

**Figure supplement 3**. Terrein determination in siderophor deletion mutants.

mutant showed a reduced growth rate without iron supplementation (*Figure 7A*), which is in agreement with reduced coprogen production as shown above (*Figure 6A*). However, severe iron limitation from the addition of the iron chelator bathophenanthroline sulfonate (BPS) completely repressed growth of the *hapX* mutant (*Figure 7A*). The ΔsidA mutant displayed the most severe phenotype: while ΔsidA showed normal growth with slightly reduced conidiation in the presence of high to moderate iron concentrations (100 or 20 µM $FeCl_3$) (*Figure 7A*), growth was strongly retarded when the iron concentration was reduced to 2 µM $FeCl_3$. No growth was observed even after prolonged incubation when iron was omitted (*Figure 7A*). These phenotypes were cured when purified coprogen was externally added (*Figure 7—figure supplement 1B*). Interestingly, growth of the ΔsidA strain was also partially restored in the presence of ascorbic acid and, although to a lesser extent, by the addition of terrein (*Figure 7A*). This result is supported by previous studies on an *A. nidulans* ΔsidA mutant which was able to grow under iron limitation in the presence of ascorbic acid (*Eisendle et al., 2003*). Unexpectedly, terrein supplementation inhibited growth of an *A. fumigatus sidA* mutant, which was also true for an *A. fumigatus* wild-type strain (*Figure 7—figure supplement 2*). Subsequent analyses showed that growth of the phytopathogen *Fusarium graminearum* was also inhibited by terrein-containing culture extracts from *A. terreus* (*Figure 7—figure supplement 2*), indicating some antifungal properties of terrein against environmental competitors.

Due to the beneficial effect of terrein on the *A. terreus* ΔsidA mutant, we investigated the impact of intrinsic terrein production on growth under iron limitation. For this, we deleted the *sidA* gene in the ΔterA background and compared growth of wild-type, ΔsidA, and ΔterAΔsidA under iron-supplemented (20 and 200 µM $FeCl_3$) and iron-limited conditions (0 and 2 µM $FeCl_3$) (*Figure 7B*). Both mutants (ΔsidA and ΔterAΔsidA) were unable to grow on media containing 2 µM or less $FeCl_3$. However, since terrein is only produced by vegetative mycelium, strains were pre-grown in the presence of 200 µM $FeCl_3$, washed, and transferred to the plates with different iron contents (*Figure 7C*). While growth of mycelium from the ΔakuB strain looked identical to that from conidia, the ΔsidA mutant started to form small colonies even in the absence of iron, whereas the ΔterAΔsidA mutant showed some weak growth only at 2 µM $FeCl_3$ but not in the absence of external iron supplementation. Terrein production was analysed from extracted agar plugs, which confirmed that the wild-type and ΔsidA mutant produced substantial amounts of terrein under iron-limited conditions whereas terrein production was fully abrogated in the ΔterAΔsidA mutant (*Figure 7—figure supplement 3*). These results indicate that terrein production supports iron acquisition, although the siderophore system is the dominant iron acquisition system.

## Discussion

*A. terreus* is known as a human pathogen that causes severe invasive bronchopulmonary and disseminated aspergillosis (*Lass-Flörl et al., 2005*; *Slesiona et al., 2012a*, *2012b*). In addition, *A. terreus* has been described as a causative agent of foliar blight of potato leaves (*Louis et al., 2013*, *2014a*, *2014b*). Besides its pathogenic potential, *A. terreus* can inhibit proliferation of other plant pathogens such as *Fusarium udum* (*Upadhyay and Rai, 1987*) and acts as a mycoparasite on sclerotia of the plant pathogenic fungus *Sclerotinia sclerotiorum* (*Melo et al., 2006*).

SMs are assumed to support competition in the plant and soil environment. The natural biological activities of terrein include the inhibition of plant seed germination (*Kamata et al., 1983*), the induction of fruit surface lesions (*Zaehle et al., 2014*), and the newly discovered reduction of ferric to ferrous iron accompanied by some antifungal activity. These activities are all appropriate to supporting the competitiveness of *A. terreus* in the environment. However, the production of terrein at the correct timing requires coordinated sensing and transduction of environmental signals.

The terrein biosynthesis gene cluster is directly controlled by the $Zn_2Cys_6$ transcription factor TerR (*Gressler et al., 2015a*). Therefore, terrein production requires transcriptional activation of *terR*. While sugars are indispensable for high terrein production rates (*Zaehle et al., 2014*), three major signals were identified that resulted in *terR* transcription: methionine-dependent induction, nitrogen limitation, and iron starvation. These signals resemble the plant and rhizosphere environment.

Plant roots are leaky for carbon compounds that derive from C-fixation during photosynthesis. Up to one third of plant-assimilated carbon may end up in the rhizosphere, and especially in the extramatrical mycelium of ectomycorrhiza (*Churchland and Grayston, 2014*). Here, at least a fraction of carbohydrates is subsequently exuded by ectomycorrhiza, making it available to other soil microorganisms (*Sun et al., 1999*). Mycorrhizas mobilize nitrogen to feed their symbiotic plant partner, resulting in

nitrogen limitation for surrounding microorganisms. These conditions stimulate terrein production in *A. terreus*, which is typically isolated from the rhizosphere (*Gao et al., 2013*; *Rajalakshmi and Mahesh, 2014*) and might affect existing plant–microbe interactions by its phytotoxic and antifungal activity (*Figure 8*).

Another inducing signal derives from methionine, which may be of particular importance when *A. terreus* acts as a potato plant pathogen (*Louis et al., 2013*). Notably, PDB induces high terrein production rates (*Zaehle et al., 2014*), which we assume to be methionine-dependent. In plants, methionine is required for ethylene production in environmental stress response (*Wang et al., 2002*), fruit ripening (*Yang and Hoffman, 1984*), and polyamine production in plant defence reactions. Furthermore, methionine is a precursor for iron chelating siderophores such as nicotianamine and mugineic acid (*Roje, 2006*) and has been identified in plant root exudates (*Dakora and Phillips, 2002*). Therefore, elevated levels of methionine depict a signal that indicates the presence of a plant environment and stimulates terrein production. However, although further studies are required to elucidate the specific contribution of terrein to plant infection, its ability to induce lesion formation on several fruit surfaces indicates that terrein interacts with plant defence mechanisms (*Zaehle et al., 2014*).

The third—at first sight unrelated—inducing factor is iron starvation. Although terrein can reduce ferric to ferrous iron, a direct positive growth-promoting effect of terrein during iron limitation in

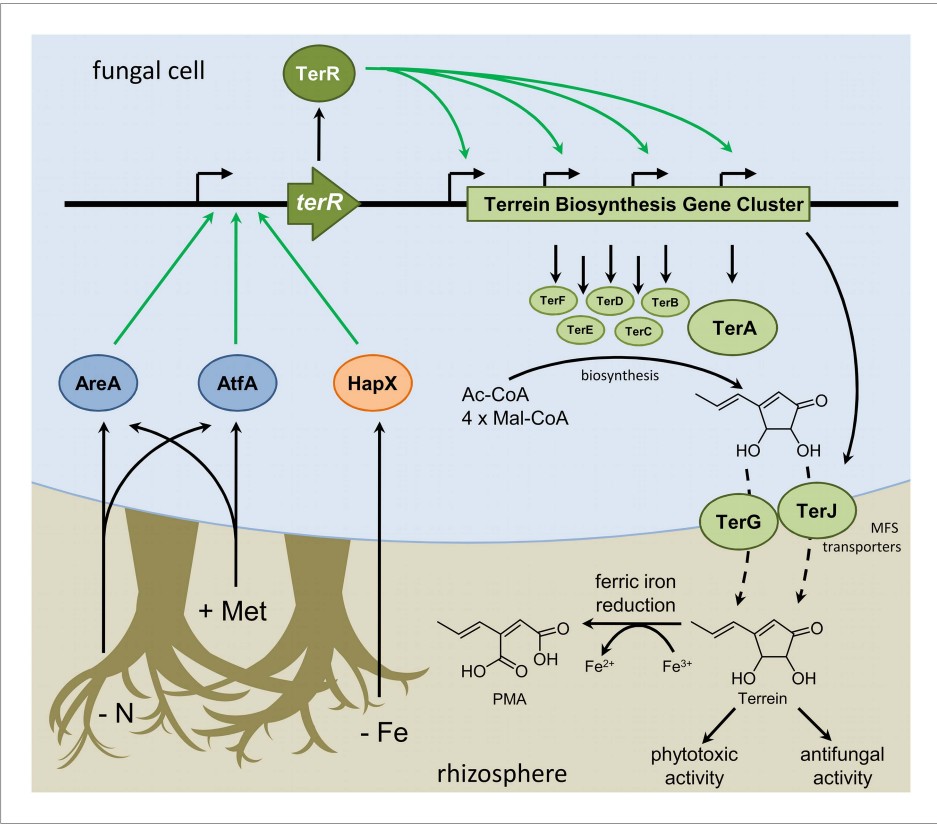

**Figure 8**. Scheme of the regulation of terrein biosynthesis gene cluster expression during interactions in the rhizosphere. Plants secrete methionine (+Met) with root exudates into the soil. Additionally, competing microorganisms reduce the available pool of nitrogen sources (−N) and iron (−Fe). Signals from nitrogen limitation and methionine are sensed via AreA and AtfA, whereas iron limitation is sensed via HapX. All three transcription factors activate the promoter of the terrein biosynthesis gene cluster-specific transcription factor. TerR leads to transcription of the structural genes required for terrein biosynthesis and terrein is produced from acetyl- and malonyl-CoA units. The two-cluster specific major facilitator superfamily (MFS) transporters export terrein into the rhizosphere. Here terrein can counteract iron limitation by its ferric iron reducing activity, supports degradation of organic matter by its phytotoxic activities, and reduces growth of competitors by its antifungal activity.

*A. terreus* monocultures was only observed in the absence of a functional siderophore system. However, extracellular siderophores show a high affinity for ferric iron but not for ferrous iron (*Hider and Kong, 2010*). Thus, the reduction of ferric iron could reduce the efficiency of iron chelation by xenosiderophores secreted from competing microorganisms. A similar strategy for iron acquisition has previously been described for the human pathogenic fungus *Histoplasma capsulatum* (*Timmerman and Woods, 1999*). *H. capsulatum* secretes glutathione accompanied by the γ-glutamyltransferase Ggt1. This enzyme releases the dipeptide cysteinylglycine from gluthathione, which in turn reduces extracellular ferric iron (*Zarnowski et al., 2008*). While *H. capsulatum* also secretes hydroxamate siderophores (*Howard et al., 2000*), silencing of *ggt1* mRNA reduced virulence in macrophages even in the presence of the siderophore system (*Zarnowski et al., 2008*). Thus, terrein may interfere with competing siderophore systems to increase the competitiveness of *A. terreus* in the environment.

In this study we showed that transduction of environmental signals to the promoter of the transcriptional activator TerR requires activation by the global transcription factors AreA, AtfA, and HapX. While AreA has previously been described to regulate several SM gene clusters in fungi (*Tudzynski et al., 1999*), HapX activates iron acquisition systems (*Schrettl et al., 2010*) and AtfA plays a specific role in osmotic and oxidative stress responses (*Lara-Rojas et al., 2011*).

In *Aspergillus* species and maybe all saprophytic ascomycetes, the GATA transcription factor AreA plays a substantial role in regulation of nitrogen. Under nitrogen starvation, AreA is transported into the nucleus (*Todd et al., 2005*) where it binds a 5′-HGATAR-3′ sequence in its target promoters (*Ravagnani et al., 1997*), as shown for the genes encoding nitrate reductase (NiaD) and nitrite reductase (NiiD) (*Chang et al., 1996*) that are upregulated in the absence of preferred nitrogen sources (*Johnstone et al., 1990*; *Punt et al., 1995*). In Aspergillus *parasiticus* AreA also binds to the promoter regions of the transcriptional regulator genes *aflR* and *aflJ* from the aflatoxin biosynthesis cluster (*Chang et al., 2000*). In contrast to nitrate utilisation, this binding impedes aflatoxin production probably due to inhibition of transcriptional activation (*Yu et al., 2004*). We have shown that terrein biosynthesis in *A. terreus* during nitrogen starvation is AreA-dependent, and at least two *areA* binding sites in the *terR* promoter were recognised by the AreA DNA-binding domain. Therefore, AreA directly binds to the *terR* promoter and supports the expression of the terrein biosynthesis genes. Similarly, biosynthesis of various mycotoxins and gibberellins is activated by AreA in phytopathogenic *Fusarium* species (*Mihlan et al., 2003*; *Kim and Woloshuk, 2008*; *Giese et al., 2013*). This implies that AreA plays a general role in regulation of SM production of plant pathogens, and the plant-derived environment may be characterised as a habitat with a high C:N ratio.

In several ascomycetes the nitrogen starvation response is also mediated by the basic leucine zipper transcription factor AtfA (*Hong et al., 2013*). In *A. nidulans*, nitrogen starvation, oxidative or osmotic stresses induce phosphorylation of the MAP kinase SakA that is subsequently transported into the nucleus to activate transcription of target genes by interaction with the constitutively expressed transcription factor AtfA (*Lara-Rojas et al., 2011*). Interestingly, in *A. terreus* the interaction between AtfA and SakA has not been solved in detail. While a *sakA* deletion does not affect terrein production levels under nitrogen limitation, it allows terrein production in the presence of some nitrogen sources, indicating that it may possess a repressor function under some conditions. In addition, a Δ*sakA* strain still produces pigmented conidia (*Figure 4—figure supplement 5*) and the *sakA* mutant—but not the Δ*atfA* strain—showed some increased osmosensitivity (*Figure 4—figure supplement 5*), which agrees with recent data on Δ*atfA* and Δ*sakA* mutants of *Penicillium marneffei* (*Nimmanee et al., 2014*). Therefore, it is conceivable that terrein biosynthesis gene cluster activation via AtfA does not require SakA.

In *A. terreus* AtfA seems to act as a general inducer of SM production since its deletion resulted in non-pigmented conidia and in the loss of terrein production under nitrogen starvation or methionine supplementation. In contrast, deletion of the *atf1* gene in *Botrytis cinerea* resulted in accumulation rather than depletion of the polyketide botcinin A and the sesquiterpene botrydial (*Temme et al., 2012*). This points to AtfA as a transcriptional inhibitor in *B. cinerea* whereas AtfA acts as an inducer in *A. terreus*. Interestingly, AreA and AtfA sense the same environmental signals to induce the terrein biosynthesis gene cluster, although their global role in cellular physiology differs significantly. This adaptation of different transcription factors towards terrein production underlines a special importance of terrein for *A. terreus* during environmental competition.

Independent from AreA and AtfA, activation of the terrein biosynthesis gene cluster was also triggered by the iron response regulator HapX under iron limitation. To overcome severe iron

starvation, microorganisms frequently produce siderophores (*Pollack and Neilands, 1970*; *Eisendle et al., 2003*; *Kreutzer and Nett, 2012*) or utilise siderophores from competing species (xenosiderophores) (*Boukhalfa and Crumbliss, 2002*). In *A. fumigatus* and *A. nidulans* siderophore biosynthesis and uptake is regulated by the opposing transcription factors HapX and SreA depending on the presence or absence of iron (*Schrettl et al., 2007*, *2008*, *2010*). Similarly, *A. terreus* produces the HapX and SreA regulated siderophores ferrichrysin and coprogen that allow efficient acquisition of iron. However, in competition with other micoorganisms, terrein-mediated iron reduction could reduce binding of iron to high-affinity siderophores from competitors that cannot be used by *A. terreus*, as described for the cysteinylglycine iron reduction system from *H. capsulatum*.

The control of terrein production by HapX depicts an unprecedented example of the activation of a SM gene cluster outside the siderophore biosynthesis system. Interestingly, terrein production is only under positive HapX control, but not under negative SreA feedback control, which is in agreement with a lack of SreA binding sites in the *terR* promoter (*Schrettl et al., 2008*), whereas a CCAAT box required for binding of the CCAAT-binding complex (CBC) and its subsequent CBC interaction with HapX (*Hortschansky et al., 2007*; *Gsaller et al., 2014*) is present. In the HapX target promoter regions of *cycA*, *sreA*, *acoA*, and *lysF* from *Aspergillus* species, a bipartite CBC–HapX-DNA binding motif with consensus 5′-C(C/A)AATCAN$_{11-12}$GAT-3′ is present, of which CCAAT is recognised by the CBC complex while the GAT sequence is bound by HapX (*Hortschansky et al., 2015*). Interestingly, these are all types of promoters that are repressed under iron limitation. In contrast, the *cccA* promoter, which encodes for a vacuolar iron transporter, is the only HapX-induced promoter known so far with a confirmed in vitro HapX binding site (*Gsaller et al., 2014*). HapX binds to the GAT motif of a 5′-CCAATN$_{16}$GATC-3′ sequence, which is also present in the *cccA* promoter of *A. terreus* and—with one mismatch—in the *terR* promoter (position −215). However, further studies will be required to confirm this motif as a HapX binding site.

In conclusion, we have shown that the terrein biosynthesis gene cluster is activated by at least three independent environmental signals: methionine, nitrogen limitation, and iron starvation. Additionally, the three global transcription factors AreA, AtfA, and HapX are essential to transfer these signals to the terrein biosynthesis gene cluster. Although one environmental signal is sufficient to induce the gene cluster, a combination of different signals provides an environmental image of higher resolution and allows an adequate adaptation to the ecological niche. The phytotoxic and antifungal potential of terrein combined with its iron reductive properties enables *A. terreus* to acquire nutrients that might otherwise be consumed by competitors. Notably, similar to terrein, the mycotoxin production in *Fusarium* species is generally associated with sensing of nitrogen availability. Although AreA plays an important regulatory function, involvement of AtfA as an additional regulator and methionine as a second inducer in these species needs to be tested. However, from our results on the terrein biosynthesis gene cluster, we propose that manifold regulation of SM gene clusters and the interplay of different global transcriptional activators might depict a general mechanism to regulate the flexibility of gene cluster activation in fungi.

## Materials and methods

### Media and cultivation conditions

All genotypes of strains used in this study are summarised in (*Figure 4—source data 1*). *A. terreus* strain SBUG844 and its derivative SBUG844Δ*akuB* that show increased rates of homologous integration served as parental strains (*Gressler et al., 2011*). If not indicated otherwise, all strains were cultivated in *Aspergillus* minimal medium (AMM; http://www.fgsc.net/methods/anidmed.html) supplemented with different carbon and nitrogen sources: AMM with 100 mM glucose (AMM-G100), with 100 mM glucose and 50 mM glutamine (AMM-G100Gln50), with 100 mM glucose, 50 mM glutamine and 10 mM methionine (AMM-G100Gln50Met10), with 1% casamino acids (AMM-CA1%). Additionally, the following complex media were used: Sabouraud (Sab, Sigma Aldrich, Taufkirchen, Germany), YPD (10 g/l yeast extract, 20 g/l peptone, 20 g/l dextrose; PDB, Sigma Aldrich), potato broth (2% potato extract, Sigma Aldrich), apple juice (pH adjusted to 6.0 with NaOH), carrot juice with honey (both Wiesgart, ALDI Nord, Jena, Germany), banana juice or peach juice (both FruchtOase, Kiberg, Germany). When required, 140 μg/ml hygromycine B (Carl Roth GmbH, Karlsruhe, Germany), 0.1 μg/ml pyrithiamine or 80 μg/ml phleomycin (both Sigma Aldrich) were added. For preparation of conidia suspensions, all strains were cultivated for 4 d at 37°C on solid 2% AMM-G50Gln10 agar plates. Conidia were

harvested by overlaying colonies with water and filtering off the suspension over 40 µm cell strainers (VWR, Darmstadt, Germany). Liquid media were generally inoculated at a final concentration of $1 \times 10^6$ conidia per ml. For the $\Delta areA$ strains, inoculation densities were generally doubled to compensate for reduced growth rates. For metabolite extraction and quantification liquid cultures in 50 ml scale were used and incubated at 30°C for 48 hr or 72 hr depending on the specific experiment. For nitrogen or iron shift experiments, strains were pre-cultivated for 40 hr in 150 ml AMM-G100Gln50 or AMM-G100Gln50 with 200 µM FeCl₃. Mycelia were harvested over sterile miracloth (Merck, Darmstadt, Germany) washed twice with AMM lacking a nitrogen (AMM-N) or iron source (AMM-Fe) and aliquots were transferred to fresh 50 ml media with or without 50 mM Gln or with or without 200 µM FeCl₃. Depending on the specific experiments, samples were analysed after 12, 24, or 48 hr of cultivation.

## Metabolite extraction, quantification, purification, and structure elucidation

Metabolites were extracted from culture broth as described previously (Gressler et al., 2011). In brief, an equal volume of ethyl acetate was added and collected after defined shaking of the mixture. The procedure was repeated once. After evaporation of the solvent, residues were taken up in 1 ml methanol each and filtered. Standard extract analyses were performed on an Agilent 1100 series HPLC-DAD system coupled with a MSD trap (Agilent Technologies, Waldbronn, Germany) operating in alternating ionisation mode. Terrein quantification was carried out from 50 ml cultures as described elsewhere (Zaehle et al., 2014). For quantification of the siderophore coprogen, the complete 50 ml culture supernatants were filtered and lyophilised to dryness. The remaining solids were extracted three times with 10 ml MeOH. The solvent from the combined organic extracts was removed under reduced pressure and residues were re-dissolved in 2 ml MeOH. The resulting slurries were filtered and the filtrates analysed by HPLC measurements. HPLC analyses were carried out on an Agilent 1260 device equipped with a quaternary pump and a UV/Vis detector (Agilent Technologies; Column: Zorbax Eclipse XDB-C8, 5 µm, 150 × 4.6 mm; flow rate 1 ml/min; eluent A: $H_2O$/0.1% HCOOH, eluent B: MeOH). The gradient started with 10% B and reached 30% B after 4 min, increased to 55% B within 10 min and reached 100% B after 2 min, where it was retained for an additional 4 min. Quantification of coprogen was performed from a calibration curve of known coprogen concentrations. For correlation of coprogen to the fungal biomass, mycelia from the cultures were dried for 48 hr at 37°C and balanced and coprogen concentrations per gram dried mycelium were calculated. All quantifications were carried out in biological triplicates and technical duplicates. Isolation of coprogen for generation of the calibration curve was performed by semi-preparative HPLC from culture supernatants of the $\Delta akuB$ and $\Delta akuB\Delta terA$ strains and fractions were collected by automatic fraction collection. Separation was carried out on a Zorbax Eclipse XDB-C8, 5 µm, 250 × 4.6 mm with a flow rate of 4.0 ml/min using $H_2O$ as eluent A and MeOH as eluent B. The gradient started with 10% B, reached 30% B after 6.5 min, increased to 55% B within 16.5 min, reached 100% B after 2 min, and was retained at 100% B for an additional 6 min. For isolation of 2-(($E$)-prop-1-en-1-yl)maleic acid, the crude product from upscaled terrein reduction assays (see below) was subjected to semi-preparative HPLC using a Zorbax Eclipse XDB-C8, 5 µm, 250 × 4.6 mm with a flow rate of 4.0 ml/min, eluent A: $H_2O$/0.1% HCOOH, eluent B: acetonitrile. The gradient started with 5% B and was held for 14 min, increased to 10% B within 9 min, increased to 100% B within 2 min where it was retained for an additional 7 min. Fractions from the new metabolite formed from ferric iron reduction were collected and evaporated resulting in a white solid which revealed a $m/z$ value of 150.0495 [M + H$^+$] by HRESI-MS that perfectly matched a calculated molecular formula of $C_7H_8O_4$ containing four double-bond equivalents. $^{13}$C-NMR measurements (*Figure 5—figure supplement 4*) revealed the presence of two carbonyl groups, one terminal methyl group, and four carbons being part of a conjugated system. Two-dimensional NMR data (*Figure 5—figure supplement 5*) and analysis of all proton coupling constants from the $^1$H-NMR spectrum (*Figure 5—figure supplement 4*) finally confirmed the structure of 2-(($E$)-prop-1-en-1-yl)maleic acid. NMR spectra were recorded on a Bruker Avance III 500 and a Bruker Avance III 600 spectrometer (Bruker BioSpin GmbH, Rheinstetten, Germany) equipped with a cryoprobe head using DMSO-d6 and methanol-d4 as solvents and internal standards.

## Fruit infection and terrein quantification from fruits

Fruit surfaces were wiped with a soft tissue saturated with 70% ethanol. Apples (ALDI, type: Tenroy Royal/Gala; Germany), bananas (type Bio, Fairverbindet; tegut, Jena, Germany), and nectarines

(tegut, type: Sweet Lady, class 1, size A; Italy) were cut lengthwise using a sterile scalpel. The resulting groove was infected with 200 µl of a conidia suspension containing $4 \times 10^7$ conidia and fruits were incubated for 7–10 d at room temperature in a humid chamber. Cut but uninfected fruits served as controls. For cultivation in the presence of high levels of nitrogen, 200 µl of a 3.5 M NH$_4$Cl solution were applied prior to infection. For terrein quantification, fruits were homogenised and extracted twice with 100 ml ethyl acetate and the solid residues were collected and evaporated for dry weight determination. Terrein was quantified from extracts as described above and terrein production rates were calculated as mg terrein/g fruit dry weight. A detailed description on fruit infection, sample preparation and terrein quantification is described in Bio-protocols by *Gressler and Brock* (*2016*).

## Fruit surface spot dilution assay

To determine the induction of lesion formation on banana fruit surfaces, organic bananas (type Bio, Fairverbindet; tegut) were softly cleaned with water and air dried. 5 µl of various sequential dilutions (1:2 to 1:512) of metabolite crude extracts (dissolved in PBS) were added as a single drop to the surface. Fruits were incubated in the dark at room temperature and photographs were taken after 40 hr and 60 hr.

## Determination of ammonia in culture supernatants

The determination of ammonia was performed as described by *Weichselbaum et al.* (*1969*). From each culture 1 ml aliquots of broth were removed and centrifuged for 5 min at 16,000×*g* to remove residual mycelium. The supernatant was collected and adjusted to pH 7.0 by addition of NaOH. A 200 µl aliquot was sequentially diluted in a nitrogen-free medium and transferred to a transparent flat-bottom 96-well plate. After addition of 20 µl SC solution (6.5 g sodium salicylate (C$_7$H$_5$O$_3$Na), 6.5 g trisodium citrate (C$_6$H$_5$O$_7$Na$_3$ × 2 H$_2$O), 48.5 mg disodium pentacyanonitrosylferrate (Na$_2$Fe(CN)$_5$NO × 2 H$_2$O) in 50 ml water), the reaction was started with 20 µl DCIC solution (1.6 g NaOH, 100 mg sodium dichloroisocyanurate (C$_3$N$_3$Cl$_2$O$_3$Na) in 50 ml water). Readings were done after 4.5 hr of incubation at room temperature in a FLOUstar Omega microplate reader (BMG Labtech, Ortenberg, Germany). Plates were shaken for 1 min in a double orbital direction and absorbance at 655 nm was measured (50 flashes/well, gain 2000). Fresh growth medium served as positive and water as negative controls. A calibration curve was recorded for calculation of ammonia levels from culture broth.

## Iron reduction assay

The ferric iron-reducing assay was performed virtually as described by *Benzie and Strain* (*1996*). In brief, three different working solutions were prepared by mixing 20 parts of reagent A (0.4% sodium acetate, 1.6% acetic acid; pH 4) with 1 part of reagent B (0.4% 2,4,6-tripyridyl-S-triazine (TPTZ; Sigma) in 0.16% HCl and either 1 part of reagent C1 (88 mM–5.5 mM sequential dilutions of terrein), reagent C2 (2% ascorbic acid, serving as positive control) or reagent C3 (water, serving as negative control). From each working solution, 200 µl were transferred into wells of a 96-well plate. An iron stock solution (10 mg/ml FeCl$_3$ in 0.2 M H$_2$SO$_4$) was diluted to 0.05 mg/ml and 100 µl were added to the working solution, which started the reaction. Reduction of ferric to ferrous iron was followed by determination of the change in absorbance at 590 nm for up to 10 hr using a FLOUstar Omega microplate reader (BMG Labtech). For determination of the pH optimum of the terrein mediated reduction, reagent A was adjusted to pH values of 3.5–6.0. Reductive activity was normalised against the activity determined from ascorbic acid, which was set as 100%. All experiments were carried out in triplicate. Upscaling of the iron reduction assays for metabolite extraction was performed by incubating 20 mg (125 µmol) of terrein in a 1:2 molar ratio with either FeSO$_4$, FeCl$_3$, CH$_3$COOCu, or (CH$_3$COO)$_2$Cu in 10 ml of reagent A. Samples were incubated for 48 hr at room temperature under continuous vertical rotation (20 rpm). Metabolites were extracted three times with ethyl acetate (10 ml each). The solvent was evaporated and the residues dissolved in MeOH and analysed as described above.

## Chrome azol S (CAS) agar plate assays

Chrome azol S agar plate assays were prepared as described by *Milagres et al.* (*1999*). Holes of 8 mm diameter were punched into plates and inoculated with 100 µl methanolic solutions of coprogen, terrein, or propylene maleic acid in concentrations of 2.50, 1.25, 0.625, and 0.3125 mg/ml. Plates were incubated for 48 hr at 30°C prior to photography.

## RNA isolation, cDNA synthesis, and quantitative real-time PCR (qRT-PCR)

Mycelia from specified cultivations were briefly washed with water and ground under liquid nitrogen. RNA was isolated using the RiboPure RNA Purification Kit (Ambion, Life Technologies, Darmstadt, Germany). Residual genomic DNA (gDNA) was removed by the DNA-free kit (Ambion). cDNA was synthesised by Revert Aid Reverse transcriptase (Thermo Scientific, Schwerte, Germany) using anchored oligodT primers. qRT-PCR was carried out on a CFX384 Touch Real-Time PCR Detection System (BioRad, Munich, Germany) using the EvaGreen 5 × QPCR (ROX) Mix (Bio & Sell, Feucht, Germany) following the manufacturer's protocol and using 1:5 and 1:10 dilutions of cDNA samples serving as templates. The actin gene (actA, ATEG_06973) and the enolase gene (enoA, ATEG_02902) were used for normalisation of transcript levels yielding similar results. Normalised transcript levels were calculated as fold expression = $2^{\Delta(\text{reference} - \text{target})}$. Primers used for qRT-PCRs showed a primer efficiency of 1.89–2.0 and are listed in *Figure 4—source data 2* (*Figure 4—source data 2*).

## Cloning strategies

In general, all PCR amplifications were performed using the Phusion DNA Polymerase (Thermo Scientific). Plasmids were amplified in *E. coli* DH5α. A detailed description of all cloning procedures is given below. Oligonucleotides used for plasmid construction are listed in the *Figure 4—source data 2* (*Figure 4—source data 2*). Fungal transformation was performed as described previously (*Gressler et al., 2011*) and transformants were checked by diagnostic PCR and Southern blot analyses (see *Figure 4—figure supplement 4*).

## Generation of P*terA*:*lacZ* reporter strains

The promoter of the *terA* gene (P*terA*; 1220 bp) was amplified with oligonucleotides P47/48 from gDNA of *A. terreus* SBUG844 and ligated into the *Not*I/*Bam*HI digested plasmid *lacZ*:*trpC*T-pJET1.2 (*Gressler et al., 2011*) containing the *E. coli lacZ* gene and the *trpC* terminator sequence. The plasmid was linearised by *Not*I digestion and the *Not*I-excised *ptrA* cassette from plasmid *ptrA*-pJET (*Fleck and Brock, 2010*) was inserted. The resulting plasmid was used for transformation of *A. terreus* SBUG844 wild-type resulting in SBUG844_P*terA*:*lacZ*.

## Overexpression of the *atfA* gene

Constitutive strong expression of the *atfA* gene was performed by using the strong constitutive *gpdA* promoter from *A. nidulans* FGSC A4 (AnP*gpdA*, 1387 bp; oligonucleotides P49/50) to control expression of *A. terreus atfA*. The *atfA* ORF together with its natural terminator was amplified from gDNA of *A. terreus* SBUG844 (*atfA* + *atfA*^T, 2466 bp) using oligonucleotides P51/52. Both fragments were fused by in vitro recombination with the *Spe*I- restricted *hph*-pCRIV vector using the InFusion HD cloning kit (Clonetech Laboratories, Saint-Germain-en-Laye, France). The resulting plasmid AnP*gpdA*:*atfA*:*atfA*^T_*hph*-pCRIV was used for transformation of *A. terreus* SBUG844 wild-type resulting in strain SBUG844_AnP*gpdA*:*atfA*.

## Deletion of the *cpcA* and *rhbA* gene

The upstream and downstream fragments of *cpcA* (ATEG_03131) were amplified from gDNA of *A. terreus* SBUG844 using oligonucleotides P53/54 (943 bp) and P55/56 (853 bp). Similarly, the upstream and downstream flanks of *rhbA* (ATEG_09480) were amplified with oligonucleotides P57/58 (473 bp) and P59/60 (418 bp). The respective fragments were fused by in vitro recombination with the *ptrA* resistance cassette (1950 bp) from *ptrA*-pJET1 (*Fleck and Brock, 2010*) into the *Kpn*I-restricted pUC19 vector using the InFusion HD cloning kit (Clonetech Laboratories), resulting in *cpcA*up-*ptrA*-*cpcA*dn-pUC19, *rhbA*up-*ptrA*-*rhbA*dn-pUC19, and *dnmT*up-*ptrA*-*dnmT*dn-pUC19. The deletion cassettes were excised by *Kpn*I restriction and used for transformation of SBUG844Δ*akuB*.

## Deletion and complementation of the *areA* gene

The upstream and downstream flanks of *areA* (ATEG_07264) were amplified from gDNA of *A. terreus* SBUG844 with oligonucleotides P61/62 (820 bp) and P63/64 (731 bp). Fragments were fused by fusion PCR and ligated into the pJET1.2 cloning vector (Thermo Scientific). The fragment was excised with *Sma*I and subcloned into pUC19. After restriction with *Not*I, the *ptrA* resistance cassette (1950 bp) from *ptrA*-pJET1 (*Fleck and Brock, 2010*) was inserted resulting in *areA*up-*ptrA*-*areA*dn-pUC19.

The deletion cassette was excised by SmaI and used for transformation of SBUG844ΔakuB. For complementation, a fragment spanning the entire areA ORF including its promoter and terminator sequence was amplified from gDNA of the A. terreus type strain FGSC A1156 (NIH2624) using oligonucleotides P65/64 (4181 bp) and ligated into pJET1.2. gDNA from the type strain was used because its areA promoter sequence contained an additional PstI restriction site that allowed discrimination of complemented strains from the parental SBUG844ΔakuB strain. The complementation fragment was excised by XhoI restriction from plasmid A1156(areAup:areA:areAdn)-pJET1.2 and directly used for the transformation of SBUG844ΔakuBΔareA. Due to the inability of the ΔareA mutant to utilise various nitrogen sources, no additional resistance marker was required and transformants were regenerated on media containing nitrate as the sole nitrogen source. Complemented strains were checked by Southern blot analysis and an additional PstI control digest of the PCR-amplified areA upstream flank.

## Deletion and complementation of the atfA gene

The upstream and downstream fragments of atfA (ATEG_04664) were amplified from gDNA of A. terreus SBUG844 with oligonucleotides P66/67 (1102 bp) and P68/69 (890 bp) and fused by in vitro recombination with the NotI-excised ptrA resistance cassette (1950 bp) from ptrA-pJET1 (Fleck and Brock, 2010) into the KpnI-excised pUC19 vector using the InFusion HD cloning kit (Clonetech Laboratories) resulting in atfAup-ptrA-atfAdn-pUC19. The deletion cassette was excised by KpnI and used for transformation of SBUG844ΔakuB. For deletion of atfA in the ΔareA background, the ptrA resistance cassette was replaced by the phleomycin resistance (ble) cassette (NotI digest from ble-pJET1.2). The deletion cassette was excised by KpnI restriction and used for transformation of SBUG844ΔakuBΔareA. For complementation of the ΔatfA phenotype, a fragment spanning the entire atfA gene including its promoter and terminator region were amplified with oligonucleotides P66/70 (3319 bp) and an additional downstream fragment was amplified with P71/69 (859 bp). The fragments were fused by in vitro recombination with the NotI-excised ble resistance cassette (2073 bp) from ble-pJET1.2 into the KpnI-restricted pUC19 vector using the InFusion HD cloning kit (Clonetech Laboratories) resulting in atfAup:atfA:atfA$^T$_ble_atfAdn-pUC19. The complementation cassette was excised by KpnI and used for transformation of SBUG844ΔakuBΔatfA.

## Overexpression of terR in the ΔareA/ΔatfA background

The atfA flanking regions were used to integrate the terR gene under control of the A. nidulans gpdA promoter in the atfA locus, which results in an atfA deletion. The upstream and downstream flanks of atfA were amplified from gDNA of A. terreus SBUG844 using oligonucleotides P72/73 (1125 bp) and P74/75 (913 bp). Subsequently, a fragment already containing the fusion of AnPgpdA with the terR ORF including its native terminator (4405 bp) was amplified with oligonucleotides P76/77 from plasmid hph_AnPgpdA:terR-pUC19 (Gressler et al., 2015a). The plasmid hph-pCRIV (Fleck and Brock, 2010) was restricted with EcoRI to remove the hph resistance cassette and all three fragments were fused by in vitro recombination into the EcoRI site using the InFusion HD cloning kit (Clonetech Laboratories). The resulting plasmid atfAup_AnPgpdA:terR:terR$^T$_atfAdn_pCRIV was linearised with NotI and the ble resistance cassette from ble-pJET1.2 was inserted between the terR terminator and the atfA downstream region. The final plasmid was restricted with EcoRI, and the fragment atfAup_AnPgpdA:terR:terR$^T$_ble_atfAdn was used for transformation of A. terreus SBUG844ΔakuBΔareA to replace the atfA ORF with the terR overexpression construct.

## Partial deletion of hapX, complementation with AnhapX, and hapX sequence verification

Because the complete sequence information of the hapX locus was lacking at the beginning of this study, only a partial deletion of the A. terreus hapX gene was performed. Upstream and downstream flanks inside the hapX coding region (ATEG_08014) were amplified with oligonucleotides P82/83 (387 bp) and P84/85 (345 bp) from genomic DNA of SBUG844 and fused by in vitro recombination with the ptrA resistance cassette (1950 bp) from ptrA-pJET1 (Fleck and Brock, 2010) into the KpnI-restricted pUC19 vector using the InFusion HD cloning kit (Clonetech Laboratories) resulting in hapXup-ptrA-hapXdn-pUC19. The deletion construct was excised by KpnI restriction and used for transformation of SBUG844ΔakuB. For complementation of the partial hapX deletion by the A. nidulans hapX, the

upstream and downstream flanks of ATEG_08014 used for generation of the partial deletion construct were amplified with oligonucleotides P82/86 (381 bp) and P87/85 (341 bp) and the complete hapX ORF (AN8251) including its native promoter and terminator sequence was amplified from gDNA of A. nidulans FGSC A4 using oligonucleotides P88/89 (2648 bp). All fragments were fused by in vitro recombination with the ble resistance cassette (2073 bp) from ble-pJET1.2 (2073 bp) into the KpnI-restricted pUC19 vector using the InFusion HD cloning kit (Clonetech Laboratories) resulting in hapXup_PAnhapX:AnhapX:AnhapX$^T$_ble_hapXdn-pUC19. The complementation cassette was excised by KpnI and used for transformation of SBUG844ΔakuBΔhapX. To identify the complete coding region of the A. terreus hapX gene, long run PCRs from within the hapX gene and the last nucleotides of the known 5′- and 3′- borders were performed with oligonucleotides P110/111 (800 bp 5′ fragment) and P112/113 (2500 bp 3′ fragment). Bands were excised from agarose gels and cloned into the pJET1.2 cloning vector. Fragments were sequenced from both strands using the primer walking method (oligonucleotides P114/115 for both fragments and additionally P116, P117 and P78 for the 3′ fragment). Finally, RNA was isolated from iron starvation conditions, transcribed into cDNA, and the hapX ORF was amplified with gene-specific oligonucleotides 113/79 and sequenced with oligonucleotides P114/115. The complete hapX locus information was submitted to EMBL and can be found under accession number KP233834(Gressler et al., 2015b).

## Deletion and complementation of the sreA gene

The upstream and downstream flanking regions of sreA (ATEG_07714) were amplified from gDNA of A. terreus SBUG844 with oligonucleotides P90/91 (763 bp) and P92/93 (756 bp) and fused by in vitro recombination with the ptrA resistance cassette (1950 bp) from ptrA-pJET1 (Fleck and Brock, 2010) into the KpnI-restricted pUC19 vector using the InFusion HD cloning kit (Clonetech Laboratories) resulting in sreAup-ptrA-sreAdn-pUC19. The deletion cassette was excised by KpnI and used for transformation of SBUG844ΔakuB. For complementation, a fragment spanning the sreA coding region including its natural promoter and terminator sequence was amplified with oligonucleotides P94/95 (3018 bp) and additional downstream fragment with P96/97 (970 bp). The fragments were fused by in vitro recombination with the ble resistance cassette (2073 bp) from ble-pJET1.2 into the KpnI-restricted pUC19 vector using the InFusion HD cloning kit (Clonetech Laboratories) resulting in sreAup:sreA:sreA$^T$_ble_sreAdn-pUC19. The complementation cassette was excised by KpnI restriction and used for transformation of SBUG844ΔakuBΔsreA.

## Deletion and complementation of the sidA gene

The upstream and downstream fragments of the sidA gene (ATEG_06879) were amplified from gDNA of A. terreus SBUG844 using oligonucleotides P98/99 (764 bp) and P100/101 (939 bp) and fused by in vitro recombination with the ptrA resistance cassette (1950 bp) from ptrA-pJET1 (Fleck and Brock, 2010) into the KpnI-restricted pUC19 vector using the InFusion HD cloning kit (Clonetech Laboratories) resulting in sidAup-ptrA-sidAdn-pUC19. For transformation of the terA deletion mutant, the ptrA cassette was excised by NotI restriction and replaced by the ble cassette. The deletion cassette was excised by KpnI restriction and used for transformation of SBUG844ΔakuB and SBUG844ΔakuBΔterA. For complementation of strain SBUG844ΔakuBΔsidA, the sidA ORF including its promoter and terminator sequence was amplified with oligonucleotides P102/103 (2393 bp) and an additional downstream fragment was amplified with P104/105 (919 bp). The fragments were fused by in vitro recombination with the ble resistance cassette (2073 bp) from ble-pJET1.2 into the HindIII-restricted pUC19 vector using the InFusion HD cloning kit (Clonetech Laboratories) resulting in sidAup:sidA:sidA$^T$_ble_sidAdn-pUC19. The complementation cassette was excised by HindIII restriction and used for transformation of SBUG844ΔakuBΔsidA.

## Bacterial expression and purification of AreA polypeptide for SPR analysis

The coding sequence for the A. nidulans AreA DNA-binding domain (amino acids 663–797) was amplified from plasmid pGEX4T1-AreA-ZnF (Muro-Pastor et al., 2004) as BamHI-HindIII fragment using oligonucleotides P80/81 (Figure 4—source data 2). The amplified fragment was cloned into a modified pET-43.1a vector allowing the addition of a N-terminal (His)$_6$-tag and removable by a tobacco etch virus (TEV) protease (Novagen). (His)$_6$-AreA$_{663-797}$ was produced by E. coli BL21 (DE3) cells grown at 30°C in 1 l Overnight Express Instant TB Medium (Novagen, Darmstadt, Germany) in the presence of 1 mM Zn(OAc)$_2$. Cells (15–20 g wet weight) were collected by centrifugation, resuspended in 200 ml lysis buffer (50 mM NaH$_2$PO$_4$, 300 mM NaCl, 10% vol/vol glycerol, 40 mM imidazole, 5 mM

β-mercaptoethanol, 1 mM AEBSF, pH 8.0) and disrupted using an Emulsiflex C5 high pressure homogeniser (Avestin, Mannheim, Germany). Cleared cellular extract was loaded to a 25 ml Ni Sepharose FF (GE Heathcare, Freiburg, Germany) column and $(His)_6$-AreA$_{663-797}$ was eluted with 200 mM imidazole. After removal of the $(His)_6$ tag by adding 4 µg TEV protease per mg peptide and overnight incubation at room temperature, samples were adjusted to 150 mM NaCl and applied on a 40 ml CellufineSulfate (Millipore, Darmstatdt, Germany) column that was equilibrated with 50 mM $NaH_2PO_4$, 150 mM NaCl, 10% vol/vol glycerol, 5 mM β-mercaptoethanol, 10 µM $Zn(OAc)_2$, pH 7.5, followed by elution of AreA$_{663-797}$ with a gradient up to 2 M NaCl. Peak fractions were concentrated with an Amicon Ultra-15 10K centrifugal filter device and purified to homogeneity by size exclusion chromatography on a Superdex 75 prep grade column (GE Healthcare) in 20 mM HEPES, 300 mM NaCl, 5 mM β-mercaptoethanol, 10 µM $Zn(OAc)_2$, pH 7.5 as running buffer. The absolute molecular mass of AreA$_{663-797}$ was determined in series on a miniDawn TREOS static light scattering monitor and an Optilab T-rEX differential refractometer (Wyatt, Dernbach, Germany). The molar mass was calculated using ASTRA 6 software (Wyatt). AreA$_{663-797}$ was stored in 50% vol/vol glycerol at −20°C.

## SPR measurements

Real-time analyses were performed on a Biacore 2000 system (GE Healthcare) at 25°C. DNA duplexes were produced by annealing complementary oligonucleotides using a fivefold molar excess of the non-biotinylated oligonucleotide. The dsDNA was injected on flow cells of a streptavidin (Sigma)-coated CM3 sensor chip at a flow rate of 10 µl/min until the calculated amount of DNA that gives a maximum AreA$_{663-797}$ binding capacity of 100 RU were bound. AreA$_{663-797}$ was injected in running buffer (10 mM HEPES pH 7.4, 150 mM NaCl, 0.005% (vol/vol) surfactant P20, 5 mM β-mercaptoethanol and 1 µM $ZnCl_2$) at concentrations from 3.125 to 200 nM. Sample injection and dissociation times were set to 200 and 400 s at a flow rate of 30 µl/min. Regeneration was achieved with 10 mM Tris/HCl pH 7.5, 0.5 M NaCl, 1 mM EDTA and 0.005% (wt/vol) SDS for 1 min. Refractive index errors due to bulk solvent effects were corrected with responses from DNA-free flow cell 1 as well as subtracting blank injections. Kinetic raw data were processed and globally fitted with Scrubber 2.0c (BioLogic Software) using a 1:1 interaction model including a mass transport term.

## Statistical significance

All analyses in which statistical analyses were required were performed in biological triplicates with at least two technical replications. Significance was calculated by use of the Microsoft Excel 2007 software package using the Student's paired t-test with a two-tailed distribution; p values were marked as follows: $*p<0.05$, $**p<0.01$; $***p<0.001$.

## Acknowledgements

We greatly acknowledge Daniela Hildebrandt for assistance in plasmid construction, Carmen Karkowski and Elena Geib for assistance in metabolite quantification, Andrea Perner for HR-MS analyses, and Heike Heinecke for recording [1]H- and [13]C-NMR spectra of secondary metabolites. This work was supported by the German Science Foundation (DFG, BR-2216/4-1) and internal funding from the Hans Knoell Institute.

## Additional information

### Funding

| Funder | Grant reference | Author |
|---|---|---|
| Deutsche Forschungsgemeinschaft (DFG) | BR-2216/4-1 | Markus Gressler, Matthias Brock |
| Hans Knöll Institute | internal funding | Markus Gressler, Florian Meyer, Daniel Heine, Peter Hortschansky, Christian Hertweck, Matthias Brock |

The funders had no role in study design, data collection and interpretation, or the decision to submit the work for publication.

## Author contributions

MG, Conception and design, Acquisition of data, Analysis and interpretation of data, Drafting or revising the article; FM, DH, PH, Acquisition of data, Analysis and interpretation of data, Drafting or revising the article; CH, Analysis and interpretation of data, Drafting or revising the article; MB, Conception and design, Analysis and interpretation of data, Drafting or revising the article

# Additional files

## Major dataset

The following dataset was generated:

| Author(s) | Year | Dataset title | Dataset ID and/or URL | Database, license, and accessibility information |
|---|---|---|---|---|
| Gressler M, Meyer F, Heine D, Hortschansky P, Brock M | 2015 | Aspergillus terreus HapX bZIP transcriptional iron regulator (hapX) gene, promoter region and complete cds | http://www.ebi.ac.uk/ena/data/view/KP233834 | Publicly available at the European Nucleotide Archive (accession no. KP233834). |

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
