## [Decision Letter]

Thank you for submitting your work entitled “Phytotoxin production in *Aspergillus terreus* is regulated by independent environmental signals” for peer review at *eLife*. Your submission has been favorably evaluated by Ian Baldwin (Senior editor) and three reviewers, one of whom, Jon Clardy, is a member of our Board of Reviewing Editors.

The reviewers have discussed the reviews with one another and the Reviewing editor has drafted this decision to help you prepare a revised submission.

Summary:

Terrein, a well-known fungal secondary metabolite, causes lesions on fruit surfaces, inhibits plant seed germination, and provides iron to its producer. But the ways in which its production is regulated and the ecological relevance of its actions remained obscure. This study employed genetic knockout and complementation studies along with other assays to deconvolute the signaling pathways – nitrogen, iron, and methionine – involved in regulating terrein production through AreA, AtfA, and Hap X transcription factors. The released terrein reduces environmental Fe3^+^ (ferric ion) to Fe2^+^ (ferrous ion) to both allow it to be imported by the producing fungus and make it unavailable to microbial competitors relying on siderophores. The conclusions illustrate how terrein production and function allow its producer to thrive in rhizosphere.

Essential revisions:

1) The manuscript needs significant editing as it simply does not do justice to the work in its present form and is unlikely to engage the typical *eLife* reader. The Introduction needs to put the current studies in a more general context, and needs to be completely rewritten. The manuscript is much too long to engage readers through to the end. As noted, the authors did a very thorough set of experiments, but they don't all need to be described in such detail. The authors should describe the significant findings and relegate others to the supplementary information. In particular, much of the material in the iron section could be moved to the supplementary information. Finally there are significant editorial issues with English usage, particularly in the Introduction.

2) The controls found involve AreA mediated response to nitrogen starvation and AreA binding sites in the 5' of *terA* are characterized. It is shown that the response to methionine is mediated by the *A. terreus* homolog of the AtfA bZip transcription factor (TF). This is a little more mysterious because previous work with this TF in other fungi indicates a role in osmotic / oxidative stress responses via the SakA MAP kinase. The authors state that SakA is not involved in terrein synthesis. It is not obvious how methionine responsiveness might occur and the authors should indicate this more clearly. Furthermore from yeast studies the Atf1/Sko1 binding site is TGACGTCA – does this motif occur in the TerR promoter?

3) The authors comprehensively show that terrein cluster expression responds to Fe starvation and that this is mediated via HapX, a well-studied TF of Fe acquisition genes. HapX acts together with the CAAT binding complex and sites are found upstream of *terR*. More recently associated HapX binding sites have been found – are these present? They further show that terrein might contribute marginally to iron acquisition in the absence of siderophores and that it is capable of reducing ferric ions, an activity that might be relevant to competition with other organisms in the rhizosphere.

4) A further signal is demonstrated – a requirement for glycolytic carbon sources – consistent with plant environments. Perhaps it should be made clear that the mechanism for this is not known.

---

## [Author Response]

*1) The manuscript needs significant editing as it simply does not do justice to the work in its present form and is unlikely to engage the typical* eLife *reader. The Introduction needs to put the current studies in a more general context, and needs to be completely rewritten. The manuscript is much too long to engage readers through to the end. As noted, the authors did a very thorough set of experiments, but they don't all need to be described in such detail. The authors should describe the significant findings and relegate others to the supplementary information. In particular, much of the material in the iron section could be moved to the supplementary information. Finally there are significant editorial issues with English usage, particularly in the Introduction*.

The Introduction has now been completely revised, putting the investigation in a much broader context. We focus now in the Introduction on the potential of secondary metabolites as pharmaceutical compounds without knowledge on the ecological and physiological impact of the metabolite for the producing organism. We hope that the English style is now acceptable.

The Results section of the main manuscript has been shortened by relocating several experimental details and supporting data to the supplemental. This also led to the reduction of total Figure in the main manuscript text that was reduced from 9 to 8 figures. Although comments below required the addition of some information that was not contained in the previous version of the manuscript, the Results section had been shortened by approximately 2 full pages. This included the “iron section”. A complete removal was not possible, because one of the biological functions of terrein is iron reduction and its regulation performs via the iron response regulator HapX. However, most details on the iron acquisition system have been relocated to the supplementary information.

*2) The controls found involve AreA mediated response to nitrogen starvation and AreA binding sites in the 5' of* terA *are characterized. It is shown that the response to methionine is mediated by the* A. terreus *homolog of the AtfA bZip transcription factor (TF). This is a little more mysterious because previous work with this TF in other fungi indicates a role in osmotic / oxidative stress responses* via *the SakA MAP kinase. The authors state that SakA is not involved in terrein synthesis. It is not obvious how methionine responsiveness might occur and the authors should indicate this more clearly. Furthermore from yeast studies the Atf1/Sko1 binding site is TGACGTCA – does this motif occur in the TerR promoter*?

We also generated a *sakA* deletion mutant in order to assess SakA/AtfA interactions. However, we did not obtain indications for a stimulating effect of SakA on AtfA. While a SakA mutant revealed an increased sensitivity against osmotic stress, this was not observed for an *atfA* mutant. The *sakA* mutant produced pigmented conidia, whereas the *atfA* mutant did not. The *sakA* mutant produced terrein under nitrogen starvation, whereas the *atfA* mutant did not. However, in terms of terrein production, a deletion of *sakA* allowed terrein production in the presence of some nitrogen sources, which prohibit terrein production in the wild type. This indicates that SakA may act as a repressor on terrein production, but whether this is direct or indirect remains to be elucidated in the future. However, we added some information in the Discussion:

“In several ascomycetes nitrogen starvation response is also mediated by the basic leucine zipper transcription factor AtfA (30). In *A. nidulans*, nitrogen starvation, oxidative or osmotic stresses induce phosphorylation of the MAP kinase SakA that is subsequently transported into the nucleus to activate transcription of target genes by interaction with the constitutively expressed transcription factor AtfA (41). Interestingly, in *A. terreus* the interaction between AtfA and SakA has not been solved in detail. While a *sakA* deletion does not affect terrein production levels under nitrogen limitation, it allows terrein production in the presence of some nitrogen sources, indicating that it may possess a repressor function under some conditions. In addition, a Δ*sakA* strain still produces pigmented conidia (Figure 4—figure supplement 5) and the *sakA* mutant, but not the Δ*atfA* strain showed some increased osmosensitivity (Figure 4—figure supplement 5), which agrees to recent data on Δ*atfA* and Δ*sakA* mutants of *Penicillium marneffei* (50). Therefore, it is well conceivable that terrein biosynthesis gene cluster activation via AtfA does not require SakA” and provided an additional supplemental figure (Figure 4—figure supplement 5).

We also analysed all promoters for putative AtfA binding sites and revised the Results section accordingly:

“Therefore, we additionally searched for putative palindromic AtfA/Sko1-binding sites (5′-TKACGTMA-3′) in the promoter regions of the cluster (53). Only one hit (TGACGTCA) was identified in the promoter of the structural gene *terC*. However, if one mismatch is allowed, there is a putative binding site at position −731 relative to the ATG start codon of *terR* (5′-TGGCGTCA-3′), but it remains speculative whether this binding site is recognised by *A. terreus* AtfA. Nevertheless, it should be mentioned that even a single half site of the suggested motif could promote transcription factor binding and promoter induction (53). Therefore, we conclude that, although direct evidence for AtfA binding at the *terR* promoter is lacking, both transcription factors seem to regulate *terR* expression.”

*3) The authors comprehensively show that terrein cluster expression responds to Fe starvation and that this is mediated* via *HapX, a well-studied TF of Fe acquisition genes. HapX acts together with the CAAT binding complex and sites are found upstream of* terR*. More recently associated HapX binding sites have been found – are these present? They further show that terrein might contribute marginally to iron acquisition in the absence of siderophores and that it is capable of reducing ferric ions, an activity that might be relevant to competition with other organisms in the rhizosphere*.

We re-analysed the *terR* promoter for putative HapX binding sites and added the following text to the Discussion:

“The control of terrein production by HapX depicts an unprecedented example for the activation of a secondary metabolite gene cluster outside the siderophore biosynthesis system. Interestingly, terrein production is only under positive HapX control, but not under negative SreA feedback-control, which is in agreement with a lack of SreA binding sites in the *terR* promoter (61), whereas a CCAAT box required for binding of the CCAAT-binding complex (CBC) and its subsequent CBC interaction with HapX (31; 25) is present. In the HapX target promoter regions of *cycA*, *sreA*, *acoA* and *lysF* from *Aspergilli* a bipartite CBC-HapX-DNA binding motif with consensus 5′-C(C/A)AATCAN_11-12_GAT-3′ is present, of which CCAAT is recognised by the CBC complex, while the GAT sequence is bound by HapX (32). Interestingly these are all types of promoters that are repressed under iron limitation. In contrast, the *cccA* promoter, which encodes for a vacuolar iron transporter, is the only known HapX induced promoter known so far with a confirmed in vitro HapX binding site (25). HapX binds to the GAT motif of a 5′-CCAATN_16_GATC-3′ sequence, which is also present in the *cccA* promoter of *A. terreus* and – with one mismatch – in the *terR* promoter (position −215). However, further studies will be required to confirm this motif as a HapX binding site.”

*4) A further signal is demonstrated – a requirement for glycolytic carbon sources – consistent with plant environments. Perhaps it should be made clear that the mechanism for this is not known*.

To make this point a bit more clear, we added the following statement to the Results section:

“However, it should be mentioned that the presence of sugars was always required, since especially hexoses from mono- and disaccharides provoked strong expression under nitrogen limited conditions (Figure 3), which is in direct contrast to the production of dihydroisoflavipucine in *A. terreus* that belongs to the class of fruit and root rot toxins. This metabolite is only produced in the strict absence of sugars and requires preferred nitrogen sources such as glutamine or asparagine for induction (24).”